# Disentangling dispersion from mean reveals true heterogeneity-diversity relationships

Cameron Pellett ✉ & Rubén Valbuena

Understanding the effect of heterogeneity is fundamental to numerous fields. In community ecology, classical theory postulates that habitat heterogeneity determines niche dimensionality and drives biodiversity. However, disparate heterogeneity-diversity relationships have been empirically observed, generating increasingly complex theoretical developments. Here we show that spurious heterogeneity-diversity relationships and subsequent theories arise as artifacts of heterogeneity measures that are mean-biased for bounded continuous variables. To solve this, we derive an alternative mean-independent measure of heterogeneity for beta and gamma distributed variables that disentangles statistical dispersion from mean. Using the mean-independent measure of heterogeneity, true monotonic positive heterogeneity-diversity relationships, consistent with classical theory, are revealed in data previously presented as evidence for both hump-shaped heterogeneity-diversity relationships and theories of an area-heterogeneity trade-off for biodiversity. This work sheds light on the source of conflicting results that have hindered understanding of heterogeneity relationships in broader ecology and numerous other fields. The mean-independent measure of heterogeneity is provided as a solution, essential for understanding true mean-independent heterogeneity relationships in wider research.

Imagine an environment (or population) whose structures or resources are dispersed heterogeneously across space (or across units). Sampling this environment would reveal a distribution with many niches and high heterogeneity, where heterogeneity can be defined by any characteristic of variability or diversity of a variable's distribution (*e.g.* statistical dispersion, extent, inequality, information entropy, *etc.*). Now consider what would occur if the mean magnitude of a variable ($x$) chosen to measure said structures and resources were to shrink, be it through leaching, degradation, decay, *etc*. The entire distribution would move down the number line, though not indefinitely: for increasingly many regions the quantity of resources and structures would reach their lower boundary, often zero. At this point, the distribution would become increasingly concentrated at the boundary, and measures of heterogeneity (variance, range, Gini, *etc*.) would change (Fig. 1a, b). The result is the entanglement (*i.e.* dependence) of measures of heterogeneity and the mean, with wide-reaching

implications for evidence and theory regarding heterogeneity and statistical dispersion in numerous fields, including economics, physics, medicine and ecology[1-6].

Consider another variable ($y$; *e.g.* richness of fungi species), that has a relationship with the mean magnitude of $x$ (*e.g.* biomass of dead matter), but not with the heterogeneity of $x$ (*e.g.* dispersion in the spatial distribution of dead matter). Despite the heterogeneity having no real effect on $y$, because changes in heterogeneity are entangled with changes in the mean, the effect of the mean on $y$ would be projected onto an apparent effect of heterogeneity on $y$. Hence, regardless of the real relationship $y$ has with heterogeneity, this entanglement inevitably results in an erroneous observed relationship between heterogeneity and $y$ (Fig. 1b, c, d). These spurious observed relationships could subsequently lead to drawing of flawed theory. For this reason, isolating and removing mean-dependence from measures of heterogeneity is essential for understanding true heterogeneity relationships.

Department of Forest Resource Management, Swedish University of Agricultural Sciences, Umeå, Sweden. ✉e-mail: cameron.pellett@slu.se

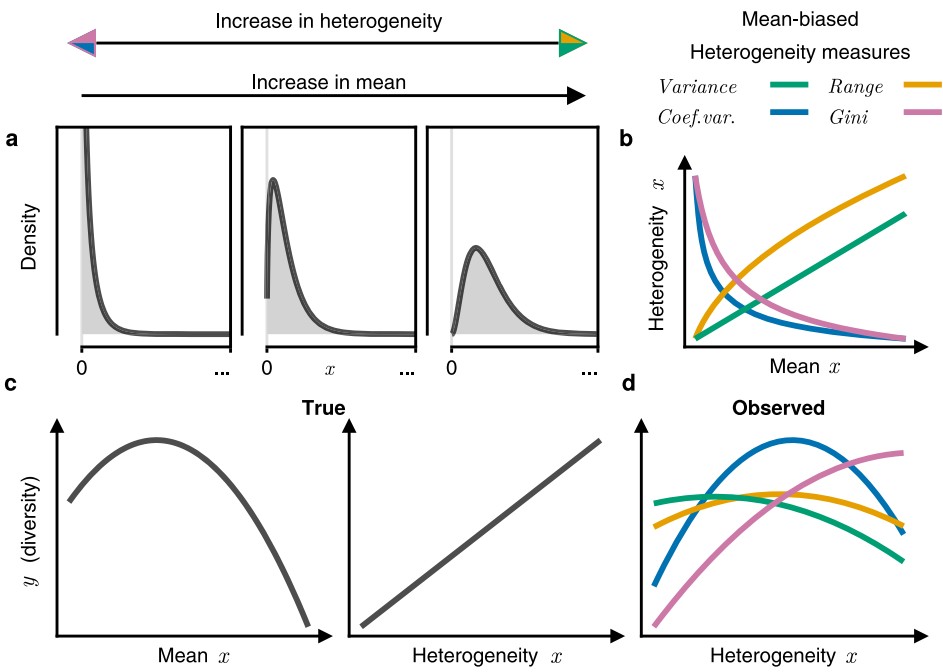

**Fig. 1 | Hypothesised entanglement of mean and heterogeneity relationships with other variables. a** As the mean of a variable ($x$; *e.g.* land elevation, foliage heights, nitrogen content, *etc.*) approaches its boundaries the distribution becomes increasingly concentrated. **b** Depending on the measure used, the observed heterogeneity has a positive or negative dependence on the mean that is often non-linear. **c** For another variable ($y$; *e.g.* bird species richness, Simpson diversity index of vascular plants, *etc.*) with any true mean-$y$ relationship and heterogeneity-$y$ relationship, using mean-biased measures of heterogeneity generates spurious observed heterogeneity-$y$ relationships (**d**) due to entanglement with the mean-$y$ relationship. Range is calculated as the extent between the 0.025 and 0.975 quartiles. The colours of the heterogeneity arrow heads for panel **a** correspond to the specific measures of heterogeneity shown in panel **b**.

Biodiversity has gained much interest in the research and political community given its global decline and its importance for resilient provision of ecosystem services[7–9]. Identification and description of factors contributing to biodiversity decline have become a priority, where the heterogeneity and mean magnitude of resources and structures in ecosystems have emerged as influential variables[10–18]. Habitat heterogeneity describes niche dimensionality and, by extension, the diversity of organisms an ecosystem can support[18]. Meanwhile, the mean magnitude of resources and structures in an ecosystem describes the available energy and niche area, which dictates the survival of coexisting species with overlapping niches[19–22]. As such, the heterogeneity and mean of an ecosystem's resources and structures are generally assumed to have ubiquitous positive relationships with biodiversity, forming the habitat heterogeneity hypothesis and energy(mean)-richness hypothesis[19,23], respectively. Contrary to classical theory, however, disparate relationships have been observed depending on the variables assessed and the measures used to describe heterogeneity[1,24–27]. This has resulted in increasingly complex developments in ecological theory arising from and attempting to explain unexpected observations that are inconsistent with classical theory.

One of the most persistent new theories, drawn to provide an explanation for observed hump-shaped heterogeneity-diversity relationships (HDRs), suggests a trade-off between the positive effects of increased heterogeneity and the opposing decrease in individual niche area that must occur with increased niche dimensionality[26–28]. The opposing decrease in individual niche area is suggested to result in fewer niche resources, reduced individual species abundances, and an increased likelihood of stochastic extinctions, eventually outweighing the positive effects of heterogeneity. Although this area-heterogeneity trade-off hypothesis could possibly explain a saturation in the capacity of the ecosystem to carry

higher diversity at extreme levels of heterogeneity, the suggested effect size at even low levels of heterogeneity when niche area is still large[27], is highly unrealistic. Unsurprisingly, the area-heterogeneity trade-off hypothesis has failed to deliver consistent evidence and controlled experiments support the contrary[24,29,30]. Crucially, the observed relationships depend on the choice of variables assessed and also on the measures used to describe heterogeneity, highlighting clear potential for the combination of mean-biases and specific mean-diversity relationships (MDRs) to together be generating spurious observed hump-shaped HDRs[1,18,29].

Here, we describe an alternative mean-independent heterogeneity measure for bounded continuous variables that disentangles statistical dispersion from the mean. We do this by showing the mean-dependence for six of the most commonly used measures of heterogeneity for continuous variables[31,32], including the coefficient of variation (CV) that is widely misconceived as mean-independent. Then, by removing the mean-dependence from the variance, we derive a measure of dispersion ($\delta$) that is unbiased by the mean, which we prove for beta and gamma distributed variables and also support with empirical data. While we can support their mean-independence with any approximately beta or gamma distributed variable, we focus on environmental variables (namely land elevation and crop cover) and on the consequences of mean-bias for theoretical and observed MDRs and HDRs. With these relationships and derived mean-independent measures of heterogeneity, we retrieve corrected HDRs from real data previously used to support the area-heterogeneity trade-off hypothesis[27,30].

## Results and discussion
### Mean-biased heterogeneity measures
Though the concept of heterogeneity is widely understood and accepted, a mean-unbiased method of quantification has not yet

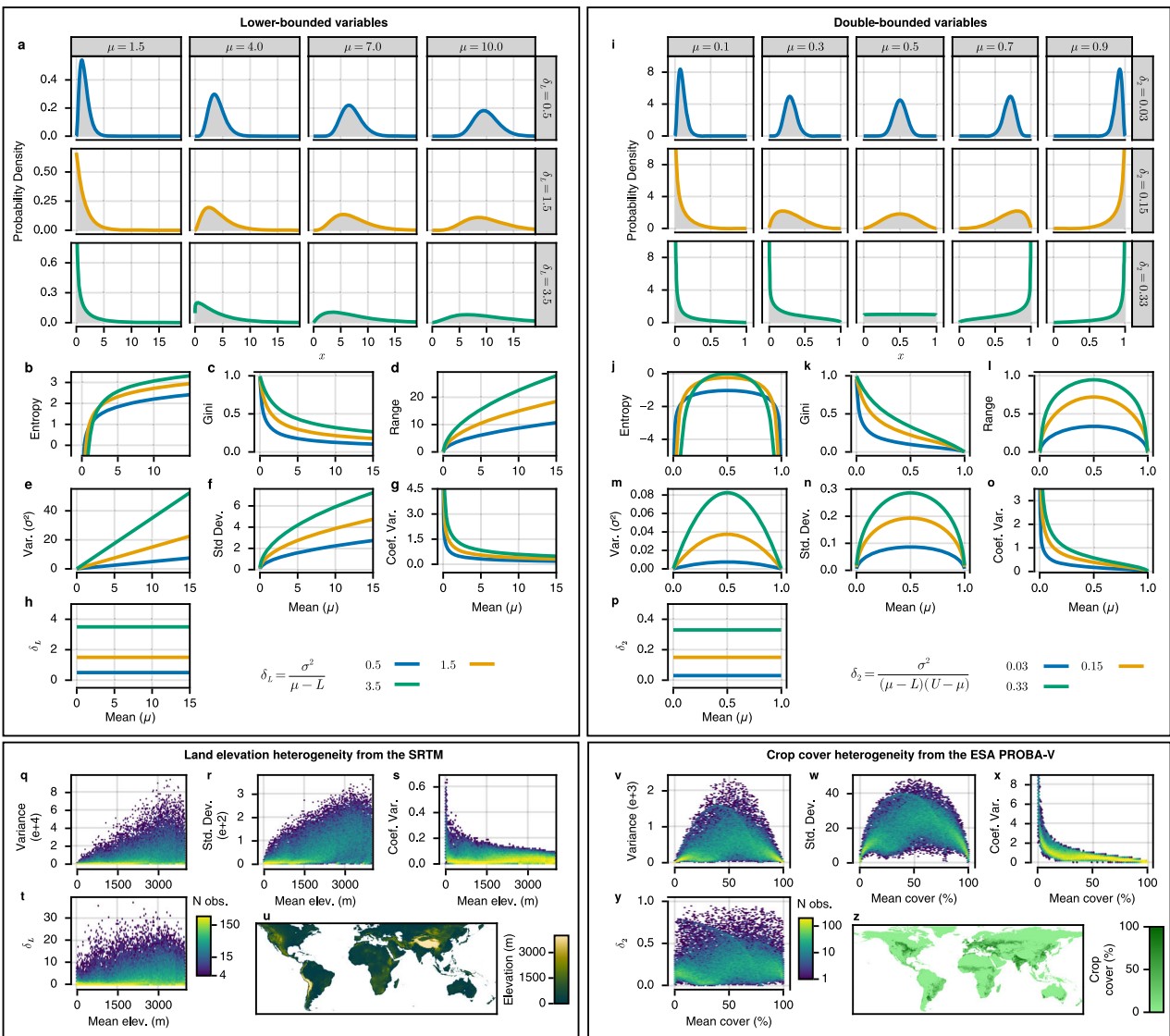

**Fig. 2 | A solution to attain mean-independence in heterogeneity measures for bounded variables. a–g, i–o** Commonly used heterogeneity measures for lower-bounded and double-bounded continuous variables are dependent on and thus inherently biased by the mean. **b–h, j–p** Heterogeneity measures are calculated based on analytically derived equations for beta and gamma distributed variables, and correspond exactly to the probability density functions (panels **a, i**). **h, p** The mean-independent measure of heterogeneity ($\delta$) can be derived by isolating and removing the mean-dependence from the variance ($\sigma^2$) (Equation (1)). **q–z** Empirical observations of heterogeneity measures for the lower-bounded variable (land elevation above sea level [$L = 0, \infty$]; 101,490 samples), and double-bounded (db)

variable (crop cover percentage [$L = 0, U = 100$]; 39,690 samples), closely match the theoretically derived relationships (Two-sided paired-sample T-test comparing theoretical models with mean model assuming no relationship; Lower-bounded: t-value $> 9.0$, df $= 101,489$, p-value $< 10^{-18}$; double-bounded: t-value $> 6.5$, df $= 39,689$, p-value $< 10^{-10}$). Land elevation above sea level is surveyed globally at a resolution of 3 arc-seconds by the shuttle radar topography mission (SRTM). Global crop cover is derived at a resolution of 3.57 arc-seconds by the Copernicus Land Monitoring Service using vegetation data collected from the ESA PROBA-V. Source data are provided for empirical panels (**q–t, v–y**) as a Source Data file.

been developed. Most current approaches for continuous variables describe characteristics of the sampled distribution, such as statistical dispersion, extent, inequality, or information entropy, where each characteristic represents a distinct and important attribute of the broader definition of heterogeneity[31,32]. Measures like the variance, range, Gini coefficient, or Shannon entropy are often used to describe these characteristics. However, the use of these measures to assess heterogeneity implies the assumption that changes in the mean simply shift the distribution up or down the number line and have no influence on its scale (statistical dispersion) or shape. This assumption is violated with the introduction of a lower and/or upper bound for a given variable. This is because populations with means further from the bounds simply have more space to be dispersed, whereas populations with means near to the bounds must

become concentrated (Fig. 2a, i), resulting in the mean-dependence and subsequent bias inherent to commonly used measures of heterogeneity (Fig. 2b–f, j–n; see methods for proof for the gamma and beta distribution). Considering a scenario in ecology, one could compare the distribution of different species along a gradient, e.g. elevation above sea level, where some species exist at a mean elevation close to zero and must have a concentrated distribution, and others exist at higher mean elevations with more space to be dispersed (the link to Rapoport's rule is discussed later in the text[33]).

Through simply dividing the standard deviation by the mean, the CV arose as an attempted solution to the mean-dependence of measures of dispersion and heterogeneity. For this reason, it is widely suggested and used without any assessment of dependence on the

mean[23,31,34]. The crucial assumption inherent to the calculation of the CV is that there is a linear dependence between the standard deviation and the mean. However, this assumption can be easily rejected when considering a variable with an upper and lower bound (double-bounded hereafter). In the case of a double-bounded variable, the standard deviation is maximised when the distribution's mean is centred between the lower and upper boundaries, because as the mean increases beyond this centre the distribution must become increasingly concentrated at the upper boundary (Fig. 2i, n). As a result, dividing the standard deviation by the mean exacerbates the measure's decrease as the mean increases beyond the centre and towards the upper boundary (Fig. 2o). The problem is furthered still when one realises the rate of change in standard deviation increases as the mean approaches the lower boundary, which influences both lower-bounded and double-bounded variables (Fig. 2f, n). Intuitively this is expected because when the mean is far from the boundaries the increased clustering of the tails of the distribution have a smaller effect on the standard deviation compared to when the mean is close to a boundary (Fig. 2a, i; see methods for rigorous definitions for gamma- and beta-distributed variables). As a result, dividing the standard deviation by the mean creates a nonlinear negative dependence between the CV and the mean for all bounded variables (Fig. 2g, o). See methods and Supplementary Fig. S1 for demonstration of the CV as a measure of skewness rather than dispersion, explaining earlier empirical results showing linear dependence between the CV and skewness[35].

## Mean-independent heterogeneity measures

To create a truly mean-independent measure of heterogeneity the exact relationship between the mean and any measure of heterogeneity must be identified and removed. We derived the variance's mean-dependence relationship for lower-bounded, double-bounded, upper bounded, and unbounded variables that are approximately beta, gamma or normally distributed, where the lower and upper bounds can be any two real numbers ($L$ and $U$, respectively). The nonlinear relationship between the standard deviation and the mean's distances from the boundaries ($\mu - L$ for the lower boundary, and $U - \mu$ for the upper boundary) can be made linear by squaring the standard deviation to give greater weight to the tails of the distribution, and retrieve the variance (Fig. 2e, m). Thus, normalising the variance ($\sigma^2$) by the mean's distances from the boundaries yields a mean-independent measure of dispersion ($\delta$). In its generalised formulation, the calculation of $\delta$ for any (bounded or unbounded) continuous variable approximated by the beta, gamma or normal distributions is given by

$$\delta = \frac{\sigma^2}{(\mu - L)^{\mathbf{1}_{\mathbb{R}}(L)}(U - \mu)^{\mathbf{1}_{\mathbb{R}}(U)}}, \qquad (1)$$

where $\mathbf{1}_{\mathbb{R}}(m)$ is an indicator function with $\mathbb{R}$ being the set of all real numbers, $m$ is the lower ($L$) or upper ($U$) bound of the variable, and $\mathbf{1}_{\mathbb{R}}(m) = 1$ if $m \in \mathbb{R}$, otherwise $\mathbf{1}_{\mathbb{R}}(m) = 0$. Note that any number raised to the zeroth power is equal to one, such that if the lower or upper bound does not exist or is not a real number (e.g. $\infty \notin \mathbb{R}$) the boundary has no influence on the dispersion nor $\delta$. Thus, specific cases arise from this generalised formula as combinations of existing or non-existing bounds. For double-bounded variables, the variance's mean-dependence is influenced simultaneously by the mean's distance from the lower and upper bounds, $[\mu - L][U - \mu]$, yielding

$$\delta_2 = \frac{\sigma^2}{(\mu - L)(U - \mu)}. \qquad (2)$$

For lower-bounded variables, the variance's relationship with the mean is linear with the distance from the lower bound, $\mu - L$ (Fig. 2e), and

thus

$$\delta_L = \frac{\sigma^2}{\mu - L}. \qquad (3)$$

For upper-bounded variables, the variance's relationship with the mean is linear with the distance from the upper bound, $U - \mu$, giving

$$\delta_U = \frac{\sigma^2}{U - \mu}. \qquad (4)$$

All of which arise as mean-independent measures of heterogeneity, as it can be demonstrated from the approximation of single-bounded and double-bounded variables respectively as gamma- and beta-distributed variables (Fig. 2h, p). Also implicitly included in the generalised formulation (Equation (1)) is the variance as a mean-independent measure of dispersion for unbounded continuous variables, as demonstrated with the normal distribution

$$\delta_0 = \sigma^2. \qquad (5)$$

Hence, $\delta$ is valid for bounded and unbounded continuous variables, where the bounds can be determined conceptually or experimentally. Bounds should not be determined as the observed minimum or maximum from a small sample without conceptual or experimental support. Examples of potential boundaries include the proportion of land covered by forest ($L = 0$ and $U = 1$), animal mass ($L = 0$ g), beetle size limited by oxygen ($L = 0$ cm and $U = 16$ cm), and biochemical fish depth limits ($L = 0$ m and $U = 8200$ m) (see Supplementary Note 1 for more details)[36-40]. Also note that $\delta$ is not scale invariant for single-bounded and unbounded variables. Therefore, variables must have common units for comparison (e.g. the same currency, or the same unit of distance).

## Applying heterogeneity measures to empirical data

After demonstrating theoretically that $\delta$ is mean-independent and that commonly used heterogeneity measures are biased by the mean, we proceeded with testing some empirical examples. We tested commonly used heterogeneity measures on global empirical datasets to evaluate correspondence between theoretical and observed mean-dependence. We assessed a single-bounded variable: land elevation above sea level, measured globally by the shuttle radar topography mission (SRTM)[41]; and a double-bounded variable: crop cover percentage, predicted globally using vegetation data collected from the European Space Agencies (ESA) PROBA-V satellite observations[42]. Datasets that covered large areas were essential to emphasise general effects and to minimise the influence of eventual feature relationships (e.g. regions with high mean forest cover having been actively managed to have low heterogeneity) that may conceal mean-heterogeneity relationships generated by the measures themselves.

Taking a mean-balanced sample of elevation ($n = 101,490$) and crop cover ($n = 39,690$), we observed empirical mean-heterogeneity relationships (Fig. 2q-z) that better fit models we derived analytically (Fig. 2a-p) than a mean model assuming no relationship with the mean (two-sided paired-sample T-test: $p$-value $< 10^{-10}$). The CV showed a nonlinear negative dependence for single- and double-bounded variables (single-bounded (sb): $t$-value $= 9.0$, df $= 101,489$, $p$-value $< 10^{-18}$; double-bounded (db): t-value $= 6.5$, df $= 39,689$, $p$-value $< 10^{-10}$). The standard deviation had a square root relationship with the mean's distance to the lower (and upper) boundary (sb: $t$-value $= 60.1$, df $= 101,489$, $p$-value $< 10^{-99}$; db: $t$-value $= 83.5$, df $= 39,689$, $p$-value $< 10^{-99}$). The variance had a linear relationship with the mean's distance to the boundaries (sb: $t$-value $= 93.7$, df $= 101,489$, $p$-value $< 10^{-99}$; db: t-value $= 95.0$, df $= 39,689$, $p$-value $< 10^{-99}$). Meanwhile, for $\delta$ the alternative hypothesis was instead that $\delta$ has a negligible relationship with

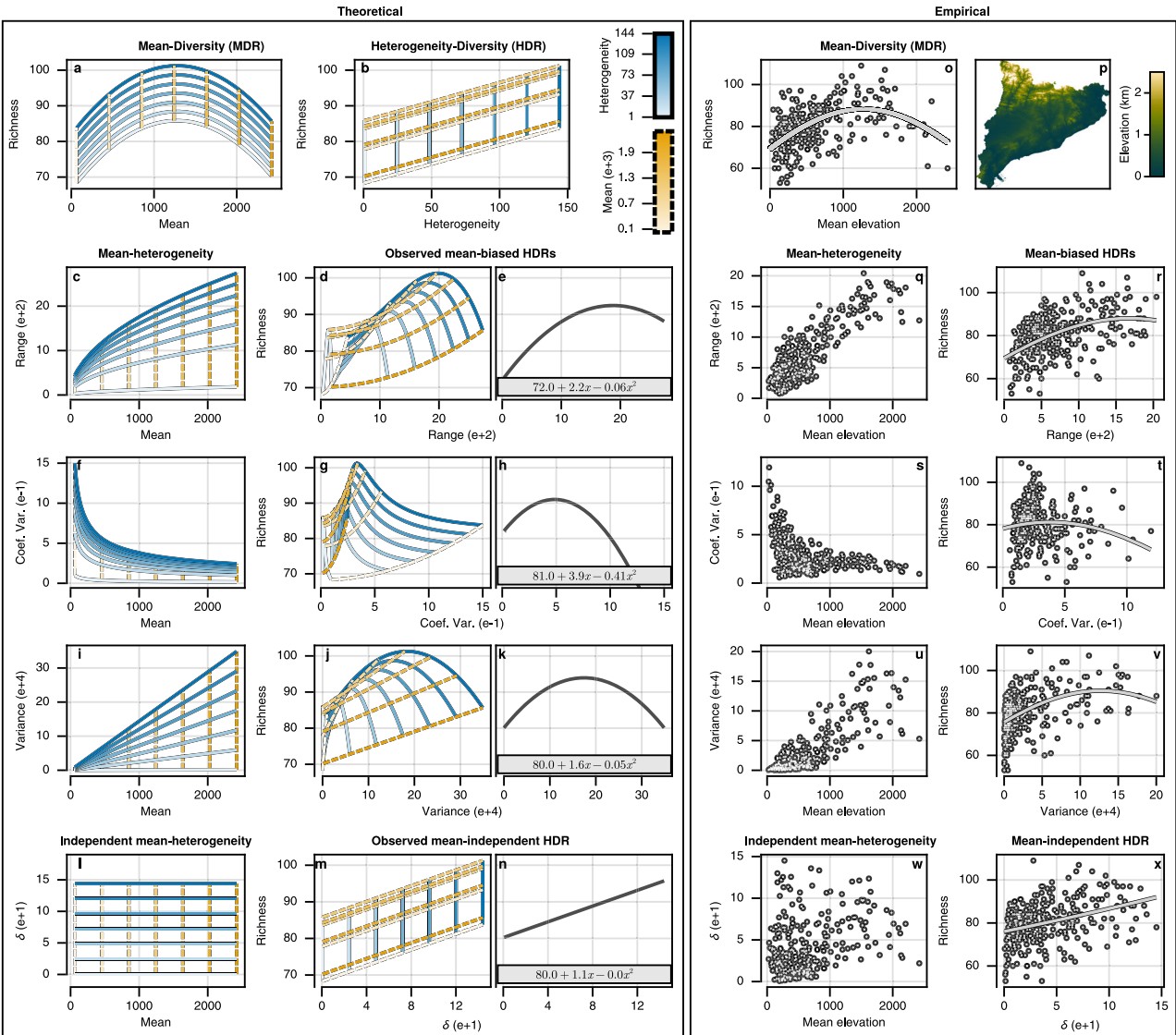

**Fig. 3 | Retrieving true heterogeneity-diversity relationships (HDRs) with a mean-independent measure of statistical dispersion ($\delta$).** We used a simplified species richness (D) model ($D = 66.395 + 0.030\mu - 0.000012\mu^2 + 0.108\delta$; Equivalent to model #5 in Table 1 and Equation (50)) where all variability is described by a hump-shaped mean relationship (**a**) and a monotonic positive heterogeneity relationship (**b**). With this model, panels **c**–**k** showed that erroneous hump-shaped HDRs arise as artifacts of mean-bias in common heterogeneity measures, where orange and blue lines were calculated from fixed levels of mean and heterogeneity, respectively. The mean-biased warping of the data resulted in erroneous hump-shaped polynomial relationships being fit (grey lines) (**e**, **h**, **k**). Mean-biased

warping of data were also observed with different species richness models (see Supplementary Figs. S2–S5). Using data for land elevation above sea level (**p**) and for breeding bird species richness in Catalonia (N = 285), empirical mean-biased HDRs were found (**q**–**v**). Alternatively, the mean-independent measure of heterogeneity ($\delta$, Equation (1)) retrieved the true underlying monotonic positive HDR in empirical data (**w**–**x**). Note that $\delta$ is implicit in the theoretical species richness model, so panels **l**–**n** are included only for comparison with the shape and concentration of empirical data. Source data are provided for empirical panels (**o**, **q**–**x**) as a Source Data file.

the mean. Testing this hypothesis against the null hypotheses of the three other model forms found for other measures of heterogeneity, we found $\delta$ showed a negligible relationship with the mean and had less than 1/3 of the absolute coefficient magnitude compared to other heterogeneity measures (sb: *t*-value > 9927, df = 101,489, *p*-value < $10^{-99}$; db: *t*-value > 3413, df = 39,689, *p*-value < $10^{-99}$), allowing analyses of heterogeneity relationships with these variables to be truly independent from the mean relationships.

## Retrieving true heterogeneity-diversity relationships

Understanding the precise effect that mean-bias has had on observed HDRs will aid in deducing robust ecological theory. When providing theory and evidence of an area-heterogeneity trade-off causing hump-shaped HDRs, Allouche et al.[27] assessed the relationship between

heterogeneity in land elevation and breeding bird species richness in Catalonia, Spain. Crucially, however, there does exist a MDR with respect to bird species richness (D) and mean land elevation above sea level[27] (Fig. 3o). The elevation MDR is expected to be hump-shaped with a negative trend due to the combination of three factors. Firstly, at high altitudes there is a reduction in available energy, resources, and plant productivity that is expected to result in a negative observed MDR (falling off rapidly at the tree line at around 2000 m to 2300 m above sea level in the Pyrenees)[27,43,44]. Secondly, anthropogenic influence has been observed in many regions to create a hump-shaped relationship driven by land use, with low elevations being dominated by croplands, mid elevations supporting diverse forests, and higher elevations becoming dominated by shrubs and conifer forests[29,30,43]. Thirdly, due to the geometric boundaries on individual species'

**Table 1 | Linear models predicting the species richness of breeding birds (D) with elevation mean and heterogeneity in Catalonia (Spain)**

| Model | Coefficient estimates (standard errors) | | | | | | | $R^2_{adj}$ |
|---|---|---|---|---|---|---|---|---|
| $D = \beta_0 + \beta_\mu \mu + ..$ | $\beta_0$ | $\beta_\mu$ | $\beta_{\mu^2}$ | $\beta_\delta$ | $\beta_{\delta^2}$ | $\beta_\rho$ | $\beta_{\rho^2}$ | |
| (#1) $..\beta_\rho \rho + \beta_{\rho^2} \rho^2$ | 70.8*** | -2.0 | | | | 41.9*** | -23.4* | 0.229 |
| | (1.5) | (5.0) | | | | (8.3) | (9.4) | |
| (#2) $..\beta_{\mu^2} \mu^2 + \beta_\rho \rho + \beta_{\rho^2} \rho^2$ | 68.5*** | 47.9*** | -64.7*** | | | 18.0* | 3.9 | 0.319 |
| | (1.4) | (9.3) | (10.4) | | | (8.7) | (9.9) | |
| (#3) $..\beta_\delta \delta + \beta_{\delta^2} \delta^2$ | 72.6*** | 12.8*** | | 21.1** | -11.6 | | | 0.194 |
| | (1.2) | (2.8) | | (7.9) | (9.3) | | | |
| (#4) $..\beta_{\mu^2} \mu^2 + \beta_\delta \delta + \beta_{\delta^2} \delta^2$ | 65.9*** | 67.2*** | -69.3*** | 23.8** | -14.9 | | | 0.327 |
| | (1.4) | (7.7) | (9.2) | (7.3) | (8.5) | | | |
| (#5) $..\beta_{\mu^2} \mu^2 + \beta_\delta \delta$ | 67.1*** | 67.5*** | -68.5*** | 11.8*** | | | | 0.322 |
| | (1.2) | (7.7) | (9.3) | (2.2) | | | | |

Models #1 and #2 used the elevation range as a measure of heterogeneity ($\rho$), following Allouche et al.[27]. Meanwhile, models #3, #4 and #5 used elevation $\delta$ as a measure of mean-independent heterogeneity (Equation (3)). Models #2, #4, and #5 account for a hump-shaped elevation mean ($\mu$) relationship. To aid comparison of effect sizes, independent variables were rescaled to values between 0 and 1 based on their observed minimum and maximum. Coefficients were estimated using ordinary least squares. All models (#1 to #5) had significantly (p-value < 0.05) smaller residuals compared to an intercept model (one-sided F-test, p-values < $10^{-15}$). Two-sided t-tests were used to evaluate significance of coefficient estimates.
$N = 285$; p-value ≤ 0.001*** 0.01** 0.05*; Intercept ($\beta_0$); Mean ($\mu$); Range ($\rho$); Dispersion ($\delta$); Richness ($D$).

distributions, a greater overlap is expected in the mid-point of the domain resulting in a greater observed species richness and a hump-shaped relationship known as the mid-domain effect[45,46]. The questions are, therefore, whether the hump-shaped MDR fully explains the observed hump-shaped HDR when using mean-biased measures of heterogeneity, and whether a mean-independent measure of heterogeneity would show a different HDR once disentangled from the MDR.

Using theoretical MDRs (Fig. 3a; panel a Supplementary Figs. S2–S5) and HDRs (Fig. 3b; panel b Supplementary Figs. S2–S5), we found that the observed HDRs when using mean-biased measures of heterogeneity are heavily biased by the MDRs (Fig. 3e, h, k; see Supplementary Figs. S2–S5 for examples with alternative predefined theoretical MDRs and HDRs). When the predefined MDR was modelled on the mean land elevation and breeding bird species richness relationship in Catalonia, we found that observed hump-shaped HDRs can arise solely as artifacts of mean-biased heterogeneity measures even when the true HDRs are monotonically positive (Fig. 3a–n). Allouche et al.[27] used the elevation range as their heterogeneity measure when providing evidence of the area-heterogeneity trade-off. The specific entanglement caused by the mean-bias of the range (Figs. 2d; 3c) results in a heavily warped observed HDR (Fig. 3d–e), which generates a polynomial relationship closely matching the one described as an area-heterogeneity trade-off [27]. Similarly, other commonly used measures of heterogeneity also fail to retrieve the true HDR underlying the model (Fig. 3f–k)[26,47].

To further support the theoretical findings, we reanalysed the SRTM land elevation and breeding bird species richness data collected in Catalonia by the Catalan Institute of Ornithology[27,29,30,41,48]. Like the theoretical model (Fig. 3a–n), we found that hump-shaped HDRs arise as artifacts of mean-biased measures of heterogeneity (Fig. 3o–v), and that the specific warping of the data matches our theoretical developments, including the fitted polynomials and expected shape and concentration of the data. When using elevation range ($\rho$) as the heterogeneity measure[27] we found that the significant (p-value < 0.01) hump-shaped relationship with species richness (Table 1, model #1) ceased to be significant after accounting for the hump-shaped MDR (Table 1, model #2). When using $\delta$ as the measure of heterogeneity, on the other hand, we only found significant (p-value < 0.01) monotonic positive relationships with species richness, with or without accounting for the hump-shaped MDR (Table 1, model #3 and #4). Moreover, we found that the inclusion of a polynomial coefficient ($\delta^2$) to assess a

hump-shaped HDR did not significantly improve the model (Table 1, model #4 and #5; F-test statistic: 3.05, df = 6, p-value = 0.08). $\delta$ elevation is unaffected by spatial auto-correlation due to having no relationship with location (Supplementary Fig. S6). Furthermore, when accounting for MDRs, using $\delta$ as a measure of elevation heterogeneity explains more variability in species richness (Table 1). Nonetheless, elevation and its heterogeneity accounted for a small portion of the variability in bird species richness (adjusted $R^2 = 0.322$), where much of the variability is likely explained by the MDR and HDR of other variables, including land use and vegetation height, cover, productivity, *etc.*

These results provide the first mean-unbiased support for the classical habitat heterogeneity hypothesis, suggesting there exists a monotonic positive HDR. We further support this with the reanalysis of forest foliage height data presented in MacArthur and MacArthur's seminal paper on HDRs[49], where we found $\delta$ retrieved a positive mean-independent HDR ($r = 0.67$, t-test statistic: 2.98, df = 11, p-value = 0.0125) without bias in manually selecting height bands for calculating Shannon entropy (coined foliage height diversity; Supplementary Fig. S7b). Conversely, we did not find evidence supporting the energy(mean)-diversity hypothesis, and no significant monotonic positive MDRs were found for foliage height ($r = 0.43$, t-statistic: 1.58, df = 11, p-value = 0.14; Supplementary Fig. S7a). These results do, however, raise uncertainty around past work, including many studies supporting positive HDRs for other continuous variables[50]. This uncertainty is raised because, for many systems, monotonic positive MDRs have been demonstrated, which will be projected onto the observed HDRs when using mean-biased measures of heterogeneity, potentially generating spurious positive relationships. Such positive MDRs have been shown for many terrestrial ecosystems where greater plant productivity, canopy height, and vegetation age correspond with increased endotherm vertebrate species richness[14,51,52], and for marine ecosystems where chemical energy (organic carbon), kinetic energy (ocean temperature) and coral biomass correlate positively with species richness of marine animals and microbes[17,23,53,54]. In all of these systems, even if changes in heterogeneity have no effect on biodiversity, the MDR would be projected onto the observed HDR when using mean-biased heterogeneity measures and spurious positive relationships would be observed (Supplementary Fig. S5). Due to this, past meta-analyses have failed to account for all biases[18], and through reanalysis is needed after accounting for the effects of MDRs. For this reanalysis, we recommend $\delta$ (Equation (1)) as the only heterogeneity

measure that finds true HDRs, unbiased by the mean for bounded continuous variables (Fig. 3l–n and w-x).

## Broader applications in ecology

Rapoport's rule suggests that the range of a species' distribution across a biogeographical gradient positively correlates with the location of that species along the gradient, where latitude[6], elevation[33], and depth[55] have been suggested as biogeographical gradients. The rule gained interest as a potential mechanism for understanding species richness gradients[56]. However, the validity of the rule has been heavily debated in the context of biogeographical gradients with bounds (e.g. edges of land masses on latitudinal gradients, summits and plateaus on elevation gradients, or the water's surface on depth gradients)[56,57]. This is because the distribution of a species must become increasingly concentrated as the mean approaches the boundaries. Therefore, even if Rapoport's rule was false and the dispersion of species along a gradient was independent of the location on the gradient, the observed relationship for the range would be expected to resemble the relationships shown in either Fig. 2d or l (depending on the number of bounds)[57]. Conversely, the $\delta$ of a species distribution should be independent of the mathematical artefacts created by the bounds, allowing the rule to be falsified or supported, such that if the rule were true, a positive correlation between $\delta$ and the mean would be present.

It is also worth noting that there are numerous distinct characteristics of heterogeneity other than dispersion, including information entropy and inequality or skewness, all with important applications. For example, diversity indexes (*e.g.* Shannon diversity index) are often used in ecology to describe the evenness in abundance between each species[49]. For which, even though we have shown that there does exist a positive correlation between dispersion ($\delta$) in foliage heights and the Shannon diversity index for bird species (Supplementary Fig. S7b), information entropy is likely a more descriptive characteristic of this specific variable due to it capturing the uniformity or measurement uncertainty of the distribution (Supplementary Fig. S7c). Likewise, measures of inequality (or skewness) are often preferred for evaluation of socio-economic outcomes where differences between all individuals are of interest[58]. Disentangling these other characteristics of heterogeneity from the mean (Fig. 2) will be an important topic of future research.

# Methods

In these methods, our objective was to demonstrate that $\delta$ is a mean-independent measure of statistical dispersion for bounded continuous variables, that other commonly used measures of heterogeneity (of which statistical dispersion is a subset) are dependent on and, thus, inherently biased by the mean, and that this bias has, in past work, resulted in spurious observed heterogeneity relationships with other variables.

## Mean-heterogeneity relationships for the standard beta and gamma distribution

Here we introduce the gamma and beta distributions for modelling lower- and double-bounded variables, respectively. For both distributions, the most frequently used heterogeneity measures have analytical derivations based solely on the parameters of the distribution, which we can use to observe mean-heterogeneity relationships without being subject to feature relationships or chance. Additionally, both distributions can be evaluated in terms of a scale/dispersion parameter that influences the spread of observations in the distribution. A parameter is interpreted as a scale/dispersion parameter when, for a constant mean, increases in the parameter always correspond to increases in the variance[59]. We can, therefore, fix the dispersion parameter, modify the mean, and observe the mean-heterogeneity relationships.

We, then, derive a method of retrieving the dispersion parameter of the gamma and beta distribution, which we denote $\delta_L$ and $\delta_2$, respectively (with the subscript referring to the type of bounds, see Equations (2) and (3)).

**Derivation of $\delta_2$ from the standard beta distribution.** For continuous variables constrained between the interval [0, 1], the beta distribution is a flexible model capable of describing a wide array of distribution shapes, including bell-shaped, left-skewed, right-skewed and multimodal (U-shaped) distributions (Fig. 2i). The beta probability density function (pdf) is typically defined by two positive shape parameters, $p$ and $q$ both on the open interval (0, ∞), with the function indexed by continuous values of $x$ constrained between the closed interval [0, 1]. The pdf for the beta distribution is

$$f(x) = x^{p-1}(1-x)^{q-1} \frac{\Gamma(p+q)}{\Gamma(p)\Gamma(q)}, \tag{6}$$

where the two exponential functions, $x^{p-1}(1-x)^{q-1}$, describe all distribution shapes modelled by the beta distribution (Fig. 2i), $\Gamma(.)$ is the gamma function, and $\Gamma(p+q)/[\Gamma(p)\Gamma(q)]$ normalises the beta density function such that the area below the function is equal to 1. Notably, as $p$ and $q$ diverge from one another and their sum remains fixed (constant), the central tendency of the distribution shifts to the left ($p < q$) or to the right ($p > q$). To be exact, the mean ($\mu$) and $1 - \mu$ can be found with

$$\mu = \frac{p}{p+q}, \tag{7}$$

and

$$1 - \mu = \frac{q}{p+q}, \tag{8}$$

where $\mu$ is constrained between the open interval (0, 1). The interval constraining $\mu$ is found with limits of the interval of $p$ and $q$ as $\lim_{p \to 0, \, q \in (0,\infty)} \mu = 0$ and $\lim_{q \to 0, \, p \in (0,\infty)} \mu = 1$. Note $\mu = 0.5$ when $p = q$. Meanwhile, as the sum of $p$ and $q$ decreases the dispersion of the distribution increases, irrespective of $\mu$. Specifically, the dispersion can be described with

$$\delta_2 = \frac{1}{1+p+q}, \tag{9}$$

where $\delta_2$ is also constrained between the open interval (0, 1). The interval constraining $\delta_2$ is found with limits of the interval of $p$ and $q$ as $\lim_{p \to \infty, \, q = (1/\mu - 1)p} \delta_2 = 0$ and $\lim_{p \to 0, \, q = (1/\mu - 1)p} \delta = 1$. Note that $\delta_2$ is equivalent to the shape parameter described by Damgaard and Irvine[60], which is better interpreted as dispersion and equivalent to the inverse of the precision parameter described by Ferrari and Cribari-Neto[59,60]. It can, thus, be shown that the combination of $\mu$ and $\delta_2$ are capable of describing all possible values of $p$ and $q$, and by extension all possible beta distributions, where

$$p = \mu(1/\delta_2 - 1) \tag{10}$$

and

$$q = (1 - \mu)(1/\delta_2 - 1). \tag{11}$$

The variance can be calculated for a beta distributed variable with

$$\sigma^2 = \frac{pq}{(p+q)^2(1+p+q)}. \tag{12}$$

However, it is clear that as $p$ and $q$ diverge from each other while keeping their sum fixed the numerator in Equation (12) will shrink, while the denominator will remain constant. Therefore, because we know $\mu$ changes as $p$ and $q$ diverge from each other, we know that the variance will simultaneously decrease as $\mu$ increases or decreases from 0.5 (Fig. 2m). This dependence between the variance and the mean is most apparent at the variance's maximum limit when $p, q \rightarrow 0$, and the variance is equal to $\mu(1 - \mu)$ (see Supplementary Note 2).

The solution to derive a mean-independent measure of dispersion for approximately beta distributed variables is to simply retrieve the distribution's dispersion parameter, $\delta_2$. Recall that the variance was dependent on the mean due to the numerator in Equation (12). However, the numerator can be cancelled out with the reciprocal of $\mu(1 - \mu) = pq/(p+q)^2$. As such the dispersion parameter of the beta distribution can be retrieved with

$$\delta_2 = \frac{\sigma^2}{\mu(1-\mu)} = \frac{1}{1+p+q}, \tag{13}$$

proving the mean independence of $\delta_2$ for beta distributed variables, where $\sigma^2$ and $\mu$ can be estimated from a sample.

**Other commonly used heterogeneity measures for the beta distribution.** The coefficient of variation (CV) is calculated for a beta distributed random variable, $X$, using Equations (7), and (12). Resulting in

$$CV(X) = \frac{\sqrt{\sigma^2}}{\mu} = \sqrt{\frac{pq}{(p+q)^2(1+p+q)}} \frac{(p+q)}{p}. \tag{14}$$

It is, however, clear from this Equation that the CV is dependent on the mean for approximately beta distributed variables. If the dispersion of the distribution, $\delta_2$, remains fixed, the components of Equation (14) that change and depend on the mean can be isolated:

$$CV(X) = \frac{\sqrt{\frac{1}{2}pq}}{p}, \qquad \text{if } p+q=1. \tag{15}$$

Like the variance, if $p$ and $q$ diverge the numerator in Equation (15) and (14) would shrink. However, unlike the variance the denominator would also change. If the divergence of $p$ and $q$ occurs due to a decreased $p$ and increased $q$ (a decrease in $\mu$) then the denominator decreases and the CV increases. Conversely, if the divergence of $p$ and $q$ occurs due to an increased $p$ and decreased $q$ (an increase in $\mu$) then the denominator increases and the CV decreases. This results in a negative nonlinear function with respect to the mean.

The Gini coefficient (G) is a frequently used measure of inequality in multiple fields, including economics and ecology. Defined as half the relative mean absolute difference, G can be calculated for a beta distributed variable, $X$, with

$$\begin{aligned} G(X) &= \frac{1}{2\mu} \int_0^\infty \int_0^\infty f(x)f(y)|x - y|dxdy \\ &= \frac{2B(2p, 2q)}{pB(p, q)^2}, \end{aligned} \tag{16}$$

where $B(.)$ is the beta function defined as $\frac{\Gamma(p)\Gamma(q)}{\Gamma(p+q)}$ (see Pham-gia and Turkkan[61] for derivation of G for the beta distribution). Though more complex to evaluate the exact relationship G has with the mean for beta distributed variables (Fig. 2k), it is clear the relationship is negative with $p$ in the denominator for the same reasons described above for the CV.

The differential entropy $H$ in nats can be calculated for a beta distributed random variable, $X$, with

$$\begin{aligned} H(X) &= -\int_0^\infty f(x) \ln f(x)dx \\ &= \ln(B(p,q)) - (p-1)[\Psi(p) - \Psi(p+q)] - (q-1)[\Psi(q) - \Psi(p+q)], \end{aligned} \tag{17}$$

where $\Psi(.)$ is the digamma function (see Lazo and Rathie[62] for derivation of H for the beta distribution). The differential entropy is highly dependent on the skewness and mean of the distribution. Although unlike other measures of heterogeneity, the differential entropy has a complex nonlinear relationship with the dispersion of the distribution. If $\mu$ is close to the (0,1) bounds, increased dispersion results in increased clustering at the lower or upper bound reducing the measured entropy, even though the expected range of observations has increased (note the crossing lines near the bounds in Fig. 2j).

The 0.95 quantile range can be approximated for a beta distributed variable using the beta distribution's cumulative distribution function (cdf),

$$F(x) = I_x(p, q), \tag{18}$$

where $I_x(.)$ is the regularized incomplete beta function. The 0.95 quantile range can then be calculated for a beta distributed variable, $X$, with

$$range(X) = x_{0.975} - x_{0.025}, \tag{19}$$

where $x_{0.975}$ and $x_{0.025}$ are the values of $x$ when $F(x)$ is equal to 0.975 and 0.025, respectively. Here we optimised $x$ using Newton's method (tolerance $= 10^{-12}$) with the initial guess set as the mode of a beta distributed variable. The resulting function calculating the range showed a similar pattern of mean-bias to the standard deviation (Fig. 2L).

**Derivation of $\delta_L$ from the standard gamma distribution.** Due to its flexibility, the standard gamma distribution has been used as a model for many zero-bounded variables in a variety of fields. The gamma distribution is commonly defined with parameters for shape ($k$) and dispersion ($\delta_L$) where the pdf, indexed by $x$, is

$$f(x) = \frac{1}{\Gamma(k)\delta_L^k} x^{k-1} e^{-x/\delta_L}, \tag{20}$$

with $x, k, \delta_L > 0$. The product of two exponential functions, $x^{k-1}e^{-x/\delta_L}$, describes all distribution shapes modelled by the pdf, while $1/(\Gamma(k)\delta_L^k)$ normalises the area below the pdf to be equal to one.

The mean and variance of the gamma distribution are, respectively,

$$\mu = k\delta_L, \tag{21}$$

and

$$\sigma^2 = k\delta_L^2 = \mu\delta_L, \tag{22}$$

where $\mu > 0$ and $\sigma^2 > 0$. It is clear from this formulation alone that the variance changes linearly with the mean while $\delta_L$ is fixed and that the rate of this change is scaled by the dispersion of the distribution (Equation (22); see Fig. 2e). It is also clear that the standard deviation ($\sigma$) has a square root relationship with the mean when the dispersion is fixed ($\sigma = \sqrt{\mu\delta_L}$).

The solution to derive a mean-independent measure of dispersion for approximately gamma distributed variables is to simply retrieve the distribution's dispersion parameter, $\delta_L$. Recall that the variance had linear dependence on the mean (Equation (22)), which can be removed with the reciprocal of $\mu$. As such, the mean-independent measure of dispersion, $\delta_L$, can be derived from the mean and variance with

$$\delta_L = \frac{\sigma^2}{\mu}, \tag{23}$$

proving the mean independence of $\delta_L$ for gamma distributed variables, where $\sigma^2$ and $\mu$ can be estimated from a sample.

**Other commonly used heterogeneity measures for the gamma distribution.** The CV can be derived from $\mu$ and $\sigma^2$ for a gamma distributed random variable, $X$, as

$$CV(X) = \frac{\sqrt{\sigma^2}}{\mu} = \frac{1}{\sqrt{k}}. \tag{24}$$

However, because the CV simplifies to be dependent solely on the shape parameter ($k$) of the gamma distribution, it better describes the mean and skewness of the distribution, instead of being descriptive of mean-independent dispersion. The skewness for a gamma distributed variable can be derived as

$$Skew(X) = \frac{2}{\sqrt{k}} = 2 \cdot CV(X). \tag{25}$$

It is noteworthy that one could fix the CV for different $\mu$, but this requires $\mu$ to be modified, counter intuitively, by the dispersion of the distribution. The result is a distribution with fixed skewness and a mode that changes little as $\mu$ increases, which would be highly unrealistic for most naturally occurring variables (Supplementary Fig. S1; Figs. 2s; 3s).

The Gini coefficient (G) can be computed for a gamma distributed random variable, $X$, with

$$G(X) = \frac{\Gamma(k+1/2)}{\sqrt{\pi}\Gamma(k+1)}. \tag{26}$$

However, like the CV, G is derived with the shape parameter of the gamma distribution, independent of the scale/dispersion parameter. As a result, G is more closely related to the skewness of the distribution, which increases when $\mu$ approaches the boundary and decreases with an increased $\mu$. This is particularly problematic for the study of developing economies for example, where G decreases as the economy becomes more wealthy on average, even if the dispersion of the wealth distribution remains constant or even increases by small amounts (see Fig. 2c). Valbuena et al.[35] elaborated on the relationship between G, CV and skew.

Here we compute the differential entropy as a measure of information. The differential entropy (H) in nats is computed for a gamma distributed random variable, $X$, with

$$H(X) = k + \ln \delta_L + \ln \Gamma(k) + (1-k)\Psi(k). \tag{27}$$

Like the differential entropy of the beta distribution there exists a complex nonlinear relationship with the mean and the dispersion. Again, increased dispersion can result in reduced entropy due to increased clustering (see crossing lines in Fig. 2b).

The 0.95 quantile range can be approximated for a gamma distributed variable using the gamma distribution's cdf,

$$F(x) = \frac{\gamma(k, x/\delta_L)}{\Gamma(k)}, \tag{28}$$

where $\gamma(.)$ is the lower incomplete gamma function. The 0.95 quantile range can then be calculated for a gamma distributed variable, $X$, following Equation (19). Here we optimise $x$ using Newton's method (tolerance $= 10^{-12}$) with the initial guess set as the mode of a gamma distributed variable (($k-1)\delta_L$ when $k \geq 1$, otherwise 0).

### Defining the general beta and gamma distribution

One limitation of the method so far is that the beta and gamma distributions are constrained to a given fixed range of values. The gamma distribution is constrained to the positive real numbers, with a zero lower bound. Meanwhile, the beta distribution is constrained to the real numbers within the interval [0, 1]. These constraints limit the variety of variables that can be modelled by the gamma and beta distributions. For this reason, here we define general variants of both distributions such that variables with any lower or upper bound can be modelled and $\delta$ can be retrieved. Rather than modifying the pdfs and cdfs directly to allow for any lower or upper bound, we defined methods for indexing the functions and for calculating the variance and mean for the distributions from observations of a variable ($V$) with any lower ($L$) or upper ($U$) bound.

**Derivation of $\delta_2$ from the general beta distribution.** Given an observation ($v$) of a random variable ($V$) with any real number upper ($U$) and lower ($L$) bound, $x$ can be retrieved for indexing the beta pdf and cdf (Equation (6); Equation (18)) with

$$x = \frac{v - L}{U - L}. \tag{29}$$

Essentially, $v$ is rescaled to a value between 0 and 1 based on its possible minimum and maximum, such that it can be used to index the beta pdf in Equation (6). Likewise, the expected value or mean of the variable, E($V$), can also be scaled to a value between 0 and 1 in the same way:

$$E(X) = \frac{E(V) - L}{U - L}, \tag{30}$$

where E($X$) is equivalent to $\mu$ for the beta distribution (defined in Equation (7)). Similarly, $1 - E(X)$, can be retrieved for $V$ with

$$1 - E(X) = \frac{U - E(V)}{U - L}. \tag{31}$$

The variance of the variable, Var($V$), scales based on the squared range between the $L$ and $U$:

$$Var(X) = \frac{Var(V)}{(U - L)^2}, \tag{32}$$

where Var($X$) is equivalent to $\sigma^2$ for the beta distribution (defined in Equation (12)). As a result, the mean-independent measure of dispersion, $\delta_2$, can be retrieved for a double-bounded variable with

$$\delta_2 = \frac{Var(V)}{(E(V) - L)(U - E(V))}. \tag{33}$$

Thus, Equation (33) is a general version of Equation (13) that can be calculated for variables with any real number lower and upper bound. In the main text Var(V) is denoted $\sigma^2$ and E(V) is denoted $\mu$ for simplicity.

The general beta pdf can be expressed using the standard beta pdf ($f(x)$; Equation (6)) as $f_g(v; L, U) = f(\frac{v-L}{U-L})/(U-L)$, where the denominator normalises the pdf to adjust for the modified scale when modelling $V$, $x$ for indexing the pdf is found for $V$ with Equation (29), and the parameters $p$ and $q$ can be found for $V$ with Equations (30), (33), (10), and (11).

**Derivation of $\delta_L$ from the general gamma distribution.** Given an observation ($v$) of a variable ($V$) with any real number lower bound ($L$), $x$ can be retrieved for indexing the gamma pdf and cdf (Equation (20); Equation (28)) with

$$x = v - L. \qquad (34)$$

Essentially $v$ is scaled such that its minimum is zero. Likewise, the expected value or mean of the variable, E(V), can be scaled such that its minimum is zero:

$$E(X) = E(V) - L, \qquad (35)$$

where E(X) is equivalent to $\mu$ for the gamma distribution (defined in Equation (21)). Conversely, the variance of the variable, Var(V), remains unchanged when converted to the variance of a gamma distributed variable:

$$Var(X) = Var(V), \qquad (36)$$

where Var(X) is equivalent to $\sigma^2$ for the gamma distribution (defined in Equation (22)). As a result, the mean-independent measure of dispersion, $\delta_L$, can be retrieved with

$$\delta_L = \frac{Var(V)}{E(V) - L}. \qquad (37)$$

Thus, Equation (37) is a general version of Equation (23) that can be calculated for variables with any real number lower bound. In the main text Var(V) is denoted $\sigma^2$ and E(V) is denoted $\mu$.

The general gamma pdf can be expressed using the standard gamma pdf ($f(x)$; Equation (20)) as $f_g(v; L) = f(v - L)$, where $x$ to be used for indexing the pdf is found for $V$ with Equation (34), and the parameters $k$ and $\delta_L$ can be found for $V$ with Equations (35), (37), and $k = \mu/\delta_L$.

**Derivation of the generalised $\delta$ formulation.** Lastly, we introduced a generalised formulation of $\delta$ that is adaptive to the number of boundaries present for the variable being measured (Equation (1)). When the upper and lower bound are real numbers ($L, U \in \mathbb{R}$), $\delta$ (Equation (1)) becomes $\delta_2$ (Equation (33)), and when the variable has only a real number lower bound ($L \in \mathbb{R}$, $U \notin \mathbb{R}$), $\delta$ (Equation (1)) becomes $\delta_L$ (Equation (37)), each with their associated proofs in previous sections. Here we provide proofs of $\delta$ for upper bounded variables and unbounded variables.

For upper bounded variables ($U \in \mathbb{R}$, $L \notin \mathbb{R}$) there exists no commonly used model distribution. The solution, however, was simply to use a conceptually flipped gamma distribution, where the lower bound of zero was instead used as an upper bound. This was done using same approach of rescaling the variable as was done for lower bounded variables. The target is, thus, to rescale a given observation ($v$) of a variable ($V$) with any real number upper bound ($U$) to retrieve $x$ for indexing the gamma pdf and cdf (Equation (20); Equation (28)). This can be done with

$$x = U - v. \qquad (38)$$

Essentially, $v$ is rescaled such that its maximum is equal to zero. Similarly, the expected value or mean of the variable (E(V)) can be scaled such that its maximum is zero:

$$E(X) = U - E(V), \qquad (39)$$

where E(X) is equivalent to $\mu$ for the gamma distribution (defined in Equation (21)). Much like the use of the general gamma distribution for lower bounded variables, the variance of the upper bounded variable (Var(V)) remains unchanged when converted to the variance of a gamma distributed variable. As a result, the mean-independent measure of dispersion for upper bounded ($U$) variables, $\delta_U$, can be retrieved with

$$\delta_U = \frac{Var(V)}{U - E(V)}. \qquad (40)$$

Thus, upper bounded variables can also be modelled using the generalised $\delta$. In the main text Var(V) is denoted $\sigma^2$ and E(V) is denoted $\mu$.

For unbounded variables ($L, U \notin \mathbb{R}$) the Gaussian/normal distribution is the most obvious model. The Gaussian distribution is defined by a parameter for central tenancy ($\mu$) and dispersion/scale ($\sigma^2$). Where both parameters are estimated independently for an unbounded variable with the mean (E(V)) and variance Var(V). Thus, the distribution's dispersion is the variance itself ($\delta_0 = Var(V) = \sigma^2$).

### Empirical data and analysis of measures of heterogeneity

We further supported the theoretical demonstration of the mean-independent $\delta$ and mean-dependence of other measures of heterogeneity by analysing two global empirical datasets as empirical examples of approximately gamma- and beta-distributed variables. As an approximately gamma-distributed variable, we analysed global land elevation above sea level measured by the shuttle radar topography mission (SRTM) at a resolution of 3 arcsec[41]. As an approximately beta-distributed variable, we analysed global crop cover predicted using vegetation data collected from the ESA PROBA-V onboard the PROBA satellite at a resolution of 3.57 arcsec[42].

Two-phase stratified random sampling was performed on the land elevation and crop cover products. The objective was not to estimate a global mean, but to simply generate a large number of samples for all possible mean values. Simple random sampling with replacement was performed first, followed by calculation of the sample means, stratification by the sample means, and resampling to generate a mean-balanced sample. Simple random sampling with replacement was performed by randomly selecting sample center positions from all pixels in each product, where each pixel had an equal probability of being selected. 10 million 60 × 60 arcsec samples were taken from the land elevation dataset, and 10 million 71.4 × 71.4 arcsec samples were collected from the crop cover product. The mean land elevation and crop cover were calculated for their respective 10 million samples each. Samples with less than 75% of their area occurring on land were removed and samples with mean land elevation above 4000 meters were removed due to a limited number of observations in many regions. Strata were defined in evenly spaced bins across the extent of the mean crop cover (0–100%) and land elevation above sea level (0-4000 meters). Stratified sampling was then performed, taking 101,490 land elevation samples in 200 uniformly spaced strata, and 39,690 crop cover samples in 50 uniformly spaced strata.

Heterogeneity measures were calculated for the mean-balanced samples of land elevation (101,490 samples) and crop cover (39,690 samples), where each sample had 400 pixels. We calculated

the sample variance ($s^2$), sample standard deviation ($s$), and coefficient of variation ($CV$). Using Equations (37) and (33), we calculated $\delta_L$ and $\delta_2$ for land elevation and crop cover, respectively. To calculate $\delta$, we used $s^2$ and the sample mean ($\overline{x}$) in place of Var($V$) and E($V$), respectively.

We assessed the fit of the models derived analytically to heterogeneity measures of empirical data, and compared the fit with a null model assuming no relationship with the mean. By calculating the mean or median observed $\delta_L$ and $\delta_2$ for all observed sample means, we could calculate the expected value of $s$, $s^2$ and $CV$ for each possible mean. For the land elevation above sea level data, with a lower bound of zero, heterogeneity measures were predicted for each sample mean value ($\overline{x_L}$) with

$$s = \sqrt{\overline{\delta_L \overline{x_L}}} + \boldsymbol{\epsilon} \qquad (41)$$

$$s^2 = \overline{\delta_L}\,\overline{x_L} + \boldsymbol{\epsilon} \qquad (42)$$

$$CV = \frac{\sqrt{\widetilde{\delta_L \overline{x_L}}}}{\overline{x_L}} + \boldsymbol{\epsilon}, \qquad (43)$$

where $\overline{\delta_L}$ is the mean $\delta_L$ observed across all samples, $\widetilde{\delta_L}$ is the median $\delta_L$ observed across all samples, and $\boldsymbol{\epsilon}$ is the residual error. For the crop cover percentage data, with a lower bound of zero and upper bound of 100, heterogeneity measures were predicted for each sample mean value ($\overline{x_2}$) with

$$s = \sqrt{\overline{\delta_2 [\overline{x_2}(100 - \overline{x_2})]}} + \boldsymbol{\epsilon} \qquad (44)$$

$$s^2 = \overline{\delta_2}[\overline{x_2}(100 - \overline{x_2})] + \boldsymbol{\epsilon} \qquad (45)$$

$$CV = \frac{\sqrt{\widetilde{\delta_2 [\overline{x_2}(100 - \overline{x_2})]}}}{\overline{x_2}} + \boldsymbol{\epsilon}, \qquad (46)$$

where $\overline{\delta_2}$ is the mean $\delta_2$ observed across all samples, $\widetilde{\delta_2}$ is the median $\delta_2$ observed across all samples, and $\boldsymbol{\epsilon}$ is the residual error. The median was deemed a better average for $\delta$ when predicting the $CV$ due to the $CV$ rapidly increasing to $\infty$ at low mean values.

Likewise, assuming no relationship with the mean, we could define a null model that is simply the mean empirically observed $s$, $s^2$ and $CV$:

$$s = \overline{s} + \boldsymbol{\epsilon} \qquad (47)$$

$$s^2 = \overline{s^2} + \boldsymbol{\epsilon} \qquad (48)$$

$$CV = \overline{CV} + \boldsymbol{\epsilon} \qquad (49)$$

where $\overline{s}$, $\overline{s^2}$ and $\overline{CV}$ are the mean of the respective heterogeneity measure across all samples, and $\boldsymbol{\epsilon}$ is the residual error. The mean squared errors of the two model approaches ($\overline{e^2}$) could then be compared for each mean-biased heterogeneity measure using a two-sided paired-sample $t$-test (relying on the central limit theorem for normality due to a large sample size). The hypothesis was that the mean $\overline{e^2}$ would be lower when accounting for the mean-bias of $s$, $s^2$ and $CV$, i.e. $\overline{e^2}$ would be lower for Equations (41-46), compared to Equations (47-49).

Regarding $\delta$, the hypothesis was that it has a negligible relationship with the mean. For this hypothesis, equivalence testing is the appropriate statistical test with previous applications in ecology[63]. The test required selecting bounds for which a coefficient would be deemed negligible, and a typical two-tailed one-sample $t$-test can be

carried out to assess confidence that the coefficient falls between the negligible bounds. We selected the negligible bounds as a absolute coefficient magnitude less than half that of all other model forms demonstrated for other measures of heterogeneity. To carry out the test, we rescaled all heterogeneity measures to values between 0 and 1 based on their observed minimum and maximum (equivalent to Equation (29)). A predictor of the heterogeneity measure was calculated based on the model forms introduced for $s$, $s^2$, and $CV$ (Equations (41-46) without the $\delta$ term). Linear models were then fit with these predictors to estimate coefficients and standard errors for both the rescaled $\delta$ and rescaled other heterogeneity measures. The t-tests were finally carried out with the coefficient estimate for $\delta$ and its standard error and compared with the null (half the absolute coefficient estimate of the associated heterogeneity measure).

We used $\delta_L$ for a variable considered to have a lower bound at zero (since terrain elevations below zero occur only at few rare exceptions), and $\delta_2$ on a variable with a lower and upper bound at 0 and 100, respectively. Nonetheless, we consider $\delta$ as a universal measure of heterogeneity and statistical dispersion that can be applied to any approximately beta or gamma distributed variable with any real number lower and upper bound. Thus many different examples of disparate bounds would be applicable, for which we give two additional examples (for more examples, see Supplementary Note 1). First, the normalised difference vegetation index (NDVI) which ranges from -1 to 1. Second, species' distributions on an elevation gradient which may have any lower and/or upper bound if there is a plateau or mountain top (note the bounds for a species' distribution on an elevation gradient are distinct from the bounds for elevation).

### Relationships between measures of heterogeneity and biodiversity

We carried out two analyses to understand the precise effect mean-bias has on observed HDRs when using mean-biased heterogeneity measures, and whether mean-independent measures of heterogeneity develop more robust ecological theory. In the first analysis, we defined theoretical species richness models that were dependent on pre-defined MDRs and HDRs, such that the observed HDRs could be calculated based on the specific mean-bias of each heterogeneity measure. In the second analysis, we used $\delta$ in a reanalysis of empirical data previously used by Allouche et al.[27] to support the area-heterogeneity trade-off hypothesis through observation of hump-shaped HDRs. We further support the empirical analysis with small-scale data for foliage height and birds species diversity[49]. With these analyses, we develop a theoretical and empirical methodology for evaluating the area-heterogeneity trade-off hypothesis that could be extended to heterogeneity relationships with any other variable in numerous fields.

**Theoretical.** Because mean-biased heterogeneity measures are produced through some mathematical combination of the mean and heterogeneity, the MDR is always projected onto the observed HDR. This is visualised in Fig. 3, and Supplementary Figs. S2–S5. In panels a and b of Fig. 3 and Supplementary Figs. S2–S5, we show the species richness for seven uniformly spaced values of mean ($\mu$) between 64 and 2422 and seven uniformly spaced values of heterogeneity ($\delta$) between 0.64 and 144.0 based on the predefined MDRs and HDRs. Using the Equations described above, the mean-biased measures of heterogeneity and mean-independent $\delta$ could be calculated for each $\mu$ and $\delta$ assuming a gamma distribution. The measure specific warping of the data could, thus, be observed for each fixed level of mean and true heterogeneity (Fig. 3d, g, j, m). For fitting a second degree polynomial to the warped data, the same process of calculating species richness was repeated with 500 uniformly spaced values of mean between 64 and 2422 and 500 uniformly spaced values of heterogeneity between 0.64 and 144.0 ($n = 500^2$). The polynomial's coefficients were fit using

ordinary least squares (Panels e, h, k, and n of Fig. 3 and Supplementary Figs. S2–S5).

For Fig. 3 we used a theoretical species richness model based on empirical observations of land elevation above sea level and its relationship with breeding bird species richness in Catalonia. The predefined MDR and HDR were, therefore, set to be equivalent to model #5 in Table 1 if the independent variables were not scaled to compare effect sizes. Meanwhile, the observed HDRs when using mean-biased heterogeneity measures were not fixed. This was done to assess if the empirical observations of hump-shaped HDRs could be attributed solely to mean-bias without influence from any other variables. For this theoretical species richness ($D$) model, all variability is described by an intercept (66.395), a hump-shaped MDR, and a monotonic positive HDR:

$$D = 66.395 + 3.034 \times 10^{-2}\boldsymbol{\mu} + -1.188 \times 10^{-5}\boldsymbol{\mu}^2 + 1.076 \times 10^{-1}\boldsymbol{\delta}. \quad (50)$$

**Empirical.** To support the area-heterogeneity trade-off hypothesis, Allouche et al.[27] assessed the relationship between the heterogeneity in land elevation above sea level and breeding bird richness data in Catalonia, Spain. Land elevation was measured by the SRTM[41] and breeding bird abundance was collected by the Institut Català d'Ornitologia for 386 10 × 10 km continuous grid cells[48]. Species richness is simply the number of unique species observed during sampling.

We made three primary changes to the analysis of the data described above. First, we calculated, for land elevation above sea level, both the range ($\rho$; used by Allouche et al.[27]) and the mean-independent $\delta$ as measures of heterogeneity. Second, we removed border grid cells that were only partially covering Catalonia and, thus, received less sampling effort, whose effect on observed species richness may not be described by a linear relationship[30]. Third, we included a polynomial of the mean elevation to account for the well described hump-shaped MDR[29]. These three changes resulted in the retrieval of a monotonic linear HDR when using a simple linear model (Table 1). Coefficients were fit using ordinary least squares. $\delta$ had no observed relationship with location (Supplementary Fig. S6). Thus, the results are unaffected by spatial auto-correlation.

To further support our results, we reanalysed the small-scale dataset used in MacArthur and MacArthur's seminal paper[49] on HDRs in forest ecosystems. The dataset consisted of 13 forests with measured bird species diversity (Shannon diversity index) and estimated foliage density across the continuous height profile. The foliage density data was digitised by sampling points along the lines and approximating the function with Chebyshev polynomials (Supplementary Fig. S7d). The approximate function was then sampled[64] to create a dataset equivalent to randomly sampling 10,000 foliage positions in each canopy. The mean and other heterogeneity measures could then be calculated as previously described. Differential entropy was approximated using k-nearest neighbour statistics[65]. We preferred differential entropy over MacArthur and MacArthur's foliage height diversity[49] due to bias induced by the apportioning of the data into height bands[66]. Pearson's correlation coefficient was subsequently calculated to assess relationships with bird species diversity.

### Reporting summary
Further information on research design is available in the Nature Portfolio Reporting Summary linked to this article.

## Data availability
The breeding bird species abundance data analysed in this study is available online in a Zenodo repository with no restrictions (https://doi.org/10.5281/zenodo.11561447)[67]. The SRTM and Copernicus land cover data are available online with no restrictions in a CGIAR-CSI and Zenodo repository, respectively (https://srtm.csi.cgiar.org/[41], and https://doi.org/10.5281/zenodo.3939050[42], respectively). Source data are provided with this paper.

## Code availability
The code for the analysis and figures of this study are available online in a Zenodo repository with no restrictions (https://doi.org/10.5281/zenodo.11561447)[67]. Analysis was carried out using the Julia (v1.10.4) programming language[68]. The full list of 303 package dependencies is given in the Zenodo repository Manifest.toml[67].

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

## Acknowledgements
We thank Jofre Carnicer and Lluís Brotons for supplying the breeding bird species abundance data for Catalonia, collected by the Institut Català d'Ornitologia. We also thank Magnus Ekström (SLU) for valuable comments on an earlier version of this manuscript.

## Author contributions
C.P.: conceptualisation, methodology, formal analysis, software, data curation, visualisation and writing of first draft. R.V.: supervision and conceptualisation. C.P. and R.V.: review and editing of the manuscript.

## Funding

## Competing interests
The authors declare no competing interests.
