## [Transparent Peer Review file · Nature Communications]

Disentangling dispersion from mean reveals true heterogeneity-diversity relationships

Corresponding Author: Mr Cameron Pellett

Version 0:

Reviewer comments:

Reviewer #1

(Remarks to the Author)

The manuscript "Disentangling dispersion from mean reveals true heterogeneity-diversity relationships" is a very valuable contribution to an important topic how to describe properly the heterogeneity of landscapes or habitats independent from the mean of resources. I am not a mathematician, but I think all assumptions are justified and I like the combination of theoretical and empirical data.

Nevertheless, I have a number of comments which might help to improve the manuscript. First, very general, I would be nice to broaden the view here somehow as removing the mean is a general challenge in many ecological studies. For instance, ecologists use standardized effect sizes to remove the effect of total number of species in a sample from the functional or phylogenetic diversity. The same applies to the relationship of observed number of species and the number of individuals collected. Even here, ecologists have used long time a $\log(\text{Species}) \sim \log(\text{Abundance})$ function to use the residuals as a measure for true species richness (after accounting for sample effort). With those examples, the authors can extend the relevance of their approach. In this context, it would be nice to discuss the findings even in light of the Rapoport's rule. The claim to have developed a statistical method for mean independent heterogeneity is well supported by the theoretical data. However, the Fig 2t suggests still an increase of heterogeneity with mean elevation. This should be evaluated statistically.

For the final analyses using the new heterogeneity and hump-shaped mean elevation I was wondering why you not test for linear effect of elevation, as surrogate for energy(mean) richness hypotheses.

Here I asked myself why you are not using additive models, which avoid a priori assumptions on the linearity of the predictor.

Line 591: Even if often misused in ecology, species richness should not be used without considering the sampling effort.

Only if you were convinced that all species are observed this would be suitable. However, this is not the case in such data as used here (see Paper by (Gotelli and Colwell 2001)). The better way would be to standardize by sample coverage, which might be a problem with the data used here (Chao and Jost 2012).

The paper is using very coarse empirical data. It would be nice to have another small-scale data set as well, e.g. local forest heterogeneity as used by the seminal paper of Robert MacArthur.

Minor points:

For Figure 1 I would recommend to explain it already using a real world measure for heterogeneity as an example to increase readability.

Line 71: needs references

Line 82-84: A link to Rapoport's rule makes sense here.

Chao, A., and L. Jost. 2012. Coverage-based rarefaction and extrapolation: standardizing samples by completeness rather than size. *Ecology* 93:2533-2547.

Gotelli, N. J., and R. K. Colwell. 2001. Quantifying biodiversity: procedures and pitfalls in the measurement and comparison of species richness. *Ecology Letters* 4: 379-391.

(Remarks on code availability)

Reviewer #2

(Remarks to the Author)

The quantification of habitat heterogeneity is not only important but also challenging. The commonly used heterogeneity measures for an environmental variable include variance, CV, Gini index and many others. Most of these indices however are dependent on the mean of the variable. This study attempts to propose a method that is independent of mean but purely measures "heterogeneity". Such a measure is useful for offering insights into the relationship between habitat heterogeneity and species diversity, particularly for testing the area-heterogeneity tradeoff hypothesis of Allouche et al. (2012). I consider this as an interesting and laudable effort. That said, I have several concerns, both conceptual and technical, for the authors to consider.

(1) Heterogeneity: To answer the question "what indices truly measure heterogeneity" really depends on the definition of heterogeneity. Without a clear definition, it is difficult to argue which methods should be favored over other methods. In this sense, methods that separate the effect of mean from heterogeneity are valuable but they do not invalidate other methods.

(2) The conceptual Fig. 1 can be further improved. It is not clear why heterogeneity increases towards the two ends of the three pdfs. Again, this really depends on how heterogeneity is defined. For example, if heterogeneity means the range of a habitat variable, then the three pdfs have the same heterogeneity (assuming they have the same x range). If heterogeneity means sampling uncertainty, then less skewed pdfs have higher heterogeneity (and uniform distribution has the highest heterogeneity).

Are the lower panels assumed or implied from the pdfs of the upper panels? I was also very confused with the term "cluster". I made me several readings to finally realize "cluster" is really "skewness" of the pdfs. Cluster is commonly used to describe spatially explicit heterogeneity, not for study like this one.

(3) Line 52: Why "in small isolated environments"? I don't see that is necessary for area-heterogeneity tradeoff.

(4) Line 78-80: This statement is misleading. For normal distribution, it is true that change in mean only shifts the distribution along the x axis. But that is not necessarily true for other distributions because mean is dependent on both shape and scale parameters, for example, in the case of Gamma distribution.

(5) Line 125: The notation of the indicator function in Equation (1) is not clear. There are no values that do not belong to real numbers \mathbb{R} . The correct notation should be $1_{\mathbb{R}}(x)=1$ if $x \in \mathbb{R}$, otherwise $1_{\mathbb{R}}(x)=0$.

(6) It seems reasonable to use the scale parameter of Gamma distribution to measure heterogeneity but I am concerned about the δ^2 of the Beta distribution (Equation 13). Let me illustrate my concern using the following two numeric cases.

Uniform distribution: When $p=q=1$, Beta distribution is a uniform distribution $U(0,1)$. In my opinion, $U(0,1)$ has the highest heterogeneity than any other Beta distribution (in terms of sampling uncertainty). This leads to $\delta^2=1/3=0.333$.

Let's now set $p=0.5, q=1$, Beta distribution becomes skewed to right (i.e., most x values are small, with mode on the smallest x). The heterogeneity of this distribution is lower than the uniform distribution, but $\delta^2=1/(1+0.5+1)=0.4$ is higher than 0.33. This is not consistent to our expectation.

(7) I am not convinced that the numeric examples shown in Fig. 3 and Fig. S2-S5 are credible. Since δ is explicitly included in Equation (50), it of course recovers the linear relationship. A real test should start with Gamma distribution (or Beta distribution) and assumes a function $y=f(x)$ that relates diversity (y) with a habitat variable (x), where x follows a Gamma (Beta) distribution. This approach does not pre-dictate any relationship between diversity and δ but let's the heterogeneity indices speak to see who would do a better job.

(8) In the section of Line 436 of page 25: In application, the lower bound (min) and upper bound (max) are unknown but subject to sampling intensity. Larger samples will have higher chance to obtain smaller min and larger max. This could be problematic in real application because two samples of different size could lead to very different δ values (I did a quick numeric calculation and saw the δ values could be 2-3 times different for different sample sizes). A solution could be to estimate min and max using order statistics.

(9) Line 406: You do not define "general beta and gamma distributions". Instead, you use the δ formulas derived from the Beta and Gamma distributions.

(10) I would suggest avoiding using "axiomatically". They are not axiomatic. You may not call δ "true" mean-independent. It is probably not for distributions other than Gamma/Beta.

(Remarks on code availability)

Version 1:

Reviewer comments:

Reviewer #1

(Remarks to the Author)

The authors did a great revision. All my comments were considered very well. I highly appreciate also the re analyses of the old MacArthur data.

(Remarks on code availability)

Reviewer #2

(Remarks to the Author)

See the attached pdf

(Remarks on code availability)

Version 2:

Reviewer comments:

Reviewer #2

(Remarks to the Author)

I thank the authors for the further clarifications. They help. The issue here is not the definition of the shape parameters of beta distribution which themselves are clear to any readers but that the ranges of the shape parameters have to be large enough. In reality, there is no theoretical basis that p and q ranges have to be large or small, nor p and q should be constrained by $p+q=\phi$. I would like the authors further to clarify that in the ms. Otherwise, I congratulate the authors for the neat contribution that is long needed.

(Remarks on code availability)

RESPONSE TO REVIEWERS' COMMENTS

1 General response

We would like to start by extended our gratitude to the reviewers for the quality of their reports. Reviewer #1 found numerous important applications of our work in broader ecology, and raised many necessary improvements to the methodology and analysed datasets. Reviewer #2 teased out numerous subtle but hugely important changes to the framing and definition of the broader concept of heterogeneity, and highlighted many needed changes to the mathematical notation and figures. Both reviewers helped greatly to improve the clarity and accuracy of the text. We think these reports have allowed us to make major improvements to the manuscript, for which we are immensely grateful.

We have responded to individual concerns and suggestions below, where we direct the reviewers to the revised manuscript with the individual line (ll.) changes. Citation numbering is identical between this peer review report and the manuscript. The reference list can be found in the manuscript.

2 Response to individual concerns

2.1 Reviewer #1

Reviewer #1 The manuscript “Disentangling dispersion from mean reveals true heterogeneity-diversity relationships” is a very valuable contribution to an important topic how to describe properly the heterogeneity of landscapes or habitats independent from the mean of resources. I am not a mathematician, but I think all assumptions are justified and I like the combination of theoretical and empirical data.

Nevertheless, I have a number of comments which might help to improve the manuscript. First, very general, I would be nice to broaden the view here somehow as removing the mean is a general challenge in many ecological studies. For instance, ecologists use standardized effect sizes to remove the effect of total number of species in a sample from the functional or phylogenetic diversity. The same applies to the relationship of observed number of species and the number of individuals collected. Even here, ecologists have used long time a $\log(\text{Species}) \sim \log(\text{Abundance})$ function to use the residuals as a measure for true species richness (after accounting for sample effort). With those examples, the authors can extend the relevance of their approach. In this context, it would be nice to discuss the findings even in light of the Rapoport’s rule.

Author response Linking this work to Rapoport’s rule is an ideal example to illustrate the utility of δ . We have, therefore, included an additional subsection (“Broader applications”) and paragraph to relate this:

(ll. 290-305) “Rapoport’s rule suggests that the range of a species’ distribution across a biogeographical gradient positively correlates with the location of that species along the gradient, where elevation, latitude, and depth have been suggested as biogeographical gradients [50, 33, 6]. The rule gained interest as a potential mechanism for understanding species richness gradients [52]. However, the validity of the rule has been heavily debated in the context of biogeographical gradients with bounds (e.g. edges of land masses on latitudinal gradients, summits and plateaus on elevation gradients, or the water’s surface on depth gradients) [51, 52]. This is because the distribution of a species must become increasingly concentrated as the mean approaches the boundaries. Therefore, even if Rapoport’s rule was false and the dispersion of species along a gradient was independent of the location on the gradient, the observed relationship for the range would be expected to resemble the relationships shown in Fig. 2d or Fig. 2l (depending on the number of bounds)[51]. Conversely, the δ of a species distribution should be independent of the mathematical artefacts created by the bounds, allowing the rule to be falsified or supported, such that if the rule were true, a positive correlation between δ and the mean would be present”.

The richness-(phylogenetic/functional)diversity and the richness-abundance relationships are also interesting examples of where disentangling dependent variables is required. Though the broad approach we used may help solving a wider set of problems, the δ we derived cannot be directly applied to these examples. This is because δ is derived for continuous variables, not discrete variables as in this case. Though there is certainly potential, extension of the ideas presented in this work to discrete cases would require substantial theoretical work, deserving of its own paper.

Reviewer #1 The claim to have developed a statistical method for mean independent heterogeneity is well supported by the theoretical data. However, the Fig 2t suggests still an increase of heterogeneity with mean elevation. This should be evaluated statistically.

Author response We absolutely agree and have included a statistical test of negligible trend. We tested all model forms found for other heterogeneity measures and included in the results that

(ll. 191-194) “we found δ showed a negligible relationship with the mean and had less than 1/3 of the absolute coefficient magnitude compared to other heterogeneity measures (sb: t-value > 9,927, df = 101,489, p-value < 10^{-99} ; db: t-value > 3,413, df = 39,689, p-value < 10^{-99})”.

The methodology for the test is included:

(ll. 620-635) “Regarding δ , the hypothesis was that it has a negligible relationship with the mean. For this hypothesis, equivalence testing is the appropriate statistical test with previous applications in ecology [58].

The test required selecting bounds for which a coefficient would be deemed negligible, and a typical two-tailed one-sample t-test can be carried out to assess confidence that the coefficient falls between the negligible bounds. We selected the negligible bounds as a absolute coefficient magnitude less than half that of all other model forms demonstrated for other measures of heterogeneity. To carry out the test, we rescaled all heterogeneity measures to values between 0 and 1 based on their observed minimum and maximum (equivalent to Equation (29)). A predictor of the heterogeneity measure was calculated based on the model forms introduced for s , s^2 , and CV (Equations (41-46) without the δ term). Linear models were then fit with these predictors to estimate coefficients and standard errors for both the rescaled δ and rescaled other heterogeneity measures. The t-tests were finally carried out with the coefficient estimate for δ and its standard error and compared with the null (half the absolute coefficient estimate of the associated heterogeneity measure)".

Reviewer #1 For the final analyses using the new heterogeneity and hump-shaped mean elevation I was wondering why you not test for linear effect of elevation, as surrogate for energy(mean) richness hypotheses.

Author response We have realised now that our justification should be better explained. We also realised that we needed to include mention of the mid-domain effect, which is frequently discussed in the literature on Rapoport's rule. Although it was not clear enough originally, we did not test a linear effect of elevation in isolation (it is tested with other variables in models #1 and #3) because

(ll. 203-204) "The elevation MDR is expected to be hump-shaped with a negative trend due to the combination of three factors".

Therefore, to test for the linear energy(mean)-richness hypotheses we would need to be able to deal with the factors that create a hump-shaped relationship for elevation-richness, which is beyond the scope of this paper. The text for the first two factors have been improved, and the third on the mid-domain effect has been added:

(ll. 204-215) "Firstly, at high altitudes there is a reduction in available energy, resources, and plant productivity that is expected to result in a negative observed MDR (falling off rapidly at the tree line at around 2,000 m to 2,300 m above sea level in the Pyrenees) [38, 39, 27]. Secondly, anthropogenic influence has been observed in many regions to create a hump-shaped relationship driven by land use, with low elevations being dominated by croplands, mid elevations supporting diverse forests, and higher elevations becoming dominated by shrubs and conifer forests [38, 30, 29]. Thirdly, due to the geometric boundaries on individual species' distributions, a greater overlap is expected

in the mid-point of the domain resulting a greater observed species richness and a hump-shaped relationship known as the mid-domain effect [40, 41]”.

However, following your suggestion of analysing additional small-scale data below, we do test for linear effects of mean foliage height, corresponding to accumulated tree biomass as a surrogate for the energy(mean)-richness hypothesis. We concluded that

- (ll. 267-270) “we did not find evidence supporting the energy(mean)-diversity hypothesis, and no significant monotonic positive MDRs were found for foliage height ($r = 0.43$, t-statistic: 1.58, $df = 11$, p-value = 0.14; Extended data Fig. S7a)”.

Reviewer #1 Here I asked myself why you are not using additive models, which avoid a priori assumptions on the linearity of the predictor.

Author response We did not use additive models (AM) because of our specific hypotheses for the model forms (testing for hump-shaped and monotonic linear relationships), which we could control for effectively with polynomial regression. AMs or generalised AM would have been a better choice for data exploration or predictive applications due to their flexibility or if we did not have specific model form to test. However, this would come with reduced interpretability, a lack of causality, and an increased likelihood of overfitting.

Reviewer #1 Line 591: Even if often misused in ecology, species richness should not be used without considering the sampling effort. Only if you were convinced that all species are observed this would be suitable. However, this is not the case in such data as used here (see Paper by (Gotelli and Colwell 2001)). The better way would be to standardize by sample coverage, which might be a problem with the data used here (Chao and Jost 2012).

Author response This is an important consideration, but this should be a minor problem for the specific data used here for two reasons [43]. Firstly, the methodology used a variable sample effort survey relying on the expertise of the surveyors to judge the time and effort required to find all species (including minority species) expected in the grid cell. Secondly, the variable effort sampling was combined with an exhaustive census of over 10% of the study area (3,281 1 x 1 km grid cells). Due to both factors the breeding bird data can be considered at least approximately standardised by completeness and large enough to minimise large errors from the true richness. There is more information on the dataset in English in Brotons et al. (2008; link). We are very grateful for the papers shared by reviewer #1, as it is now clear to us that standardising by coverage is far superior to standardising by effort. Although it cannot be applied to this dataset, we will take this on to future work.

Reviewer #1 The paper is using very coarse empirical data. It would be nice to have another small-scale data set as well, e.g. local forest heterogeneity as used by the seminal paper of Robert MacArthur.

Author response We agree with the suggestion considering δ may be a better measure of heterogeneity than the easily biased foliage height diversity (FHD). FHD requires manually selecting (three) bands for calculation, which was used by MacArthur as additional degrees of freedom to make ‘the collection of points on the graph most orderly’. We have, therefore, digitised the data presented in MacArthur’s seminal paper and included it in the manuscript as extended data with additional text in the main article to describe it:

(ll. 262-267) “We further support this with the reanalysis of forest foliage height data presented in MacArthur’s and MacArthur’s seminal paper on HDRs [44], where we found δ retrieved a positive mean-independent HDR ($r = 0.67$, t-test statistic: 2.98, $df = 11$, p-value = 0.0125) without bias in manually selecting height bands for calculating Shannon entropy (coined foliage height diversity; Extended data Fig. S7b)”.

Methods for digitising the data were also added

(ll. 707-720) “To further support our results, we reanalysed the small-scale dataset used in MacArthur’s and MacArthur’s seminal paper [44] on HDRs in forest ecosystems. The dataset consisted of 13 forests with measured bird species diversity (Shannon diversity index) and estimated foliage density across the continuous height profile. The foliage density data was digitised by sampling points along the lines and approximating the function with Chebyshev polynomials (Extended data Fig. S7d). The approximate function was then sampled [59], to create a dataset equivalent to randomly sampling 10,000 foliage positions in each canopy. The mean and other heterogeneity measures could then be calculated as previously described. Differential entropy was approximated using k-nearest neighbour statistics [60]. We preferred differential entropy over MacArthur and MacArthur’s foliage height diversity [44] due to bias induced by the apportioning of the data into height bands [61]. Pearson’s correlation coefficient was subsequently calculated to assess relationships with bird species diversity”.

Reviewer #1 Minor points: For Figure 1 I would recommend to explain it already using a real world measure for heterogeneity as an example to increase readability.

Author response We agree that describing Fig. 1 with specific real world measures would help readers. We have, therefore, added specific environmental variables in the Figure 1 description to aid interpretation:

“(x; e.g. land elevation, foliage heights, nitrogen content, etc.)” and

“(y; e.g. bird species richness, Simpson diversity index of vascular plants, etc.)”

We also added “diversity” in parentheses to the Fig. 1c y-axis. Additionally, we have added example variables to the main text to aid understanding. The sentence now reads

(ll. 19-21) “Consider another variable (y; e.g. richness of fungi species), that has a relationship with the mean magnitude of x (e.g. biomass of dead matter), but not with the heterogeneity of x (e.g. dispersion in the spatial distribution of dead matter)”.

Reviewer #1 Line 71: needs references

Author response Thanks. References to Carnicer et al. [30] and Allouche et al. [27] have been added (ll. 76).

Reviewer #1 Line 82-84: A link to Rapoport's rule makes sense here.

Author response We agree that this will be important to Rapoport's rule and a real-world example may help readers. We have, therefore, made the link to species' distributions across elevation:

(ll. 95-99) “Considering a scenario in ecology, one could compare the distribution of different species along a gradient, e.g. elevation above sea level, where some species exist at a mean elevation close to zero and must have a concentrated distribution, and others exist at higher mean elevations with more space to be dispersed (the link to Rapoport's rule is discussed later in the text [33])”.

Reviewer #1 references Chao, A., and L. Jost. 2012. Coverage-based rarefaction and extrapolation: standardizing samples by completeness rather than size. *Ecology* 93:2533-2547.

Gotelli, N. J., and R. K. Colwell. 2001. Quantifying biodiversity: procedures and pitfalls in the measurement and comparison of species richness. *Ecology Letters* 4: 379-391.

2.2 Reviewer #2

Reviewer #2 The quantification of habitat heterogeneity is not only important but also challenging. The commonly used heterogeneity measures for an environmental variable include variance, CV, Gini index and many others. Most of these indices however are dependent on the mean of the variable. This study attempts to propose a method that is independent of mean but purely measures “heterogeneity”. Such a measure is useful for offering insights into the relationship between habitat heterogeneity and species diversity, particularly for testing the area-heterogeneity tradeoff hypothesis of Allouche et al. (2012). I consider this as an interesting and laudable

effort. That said, I have several concerns, both conceptual and technical, for the authors to consider.

(1) Heterogeneity: To answer the question “what indices truly measure heterogeneity” really depends on the definition of heterogeneity. Without a clear definition, it is difficult to argue which methods should be favored over other methods. In this sense, methods that separate the effect of mean from heterogeneity are valuable but they do not invalidate other methods.

Author response We absolutely agree with the reviewer. Throughout the introduction of the text, we want the reader to consider their preferred definition or measure of heterogeneity, because they are all impacted by mean bias. This was not clear enough originally, so we have highlighted that the reader could consider heterogeneity as

- (ll. 6-8) “any characteristic of variability or diversity of a variables distribution (*e.g.* statistical dispersion, extent, inequality, information entropy, *etc.*)”.

For which we give some example measures

- (ll. 14 “(variance, range, Gini, *etc.*)”.

We have also modified the text to make it clear that

- (ll. 83-84) “each characteristic represents a distinct and important attribute of the broader definition of heterogeneity”.

Additionally, we have removed the reference to finding a

- (ll. 79) “singular”

measure of heterogeneity, because there should not be just one characteristic of heterogeneity, rather the characteristic should be selected for the application. However, to avoid the problems of mean bias introduced by this work these other characteristics will also need to be disentangled from the mean. This is an important topic of future work, particularly for information entropy, which would be complementary to the utility of the mean-independent measure of dispersion introduced here. In this work we disentangled dispersion from mean because dispersion is arguably one of the most important heterogeneity measures, and is widely used for numerous topics. See also the application to Rapoport’s rule suggested by reviewer #1 (discussion added in ll. 290-305), where dispersion would be the preferred characteristic over another like information entropy.

Reviewer #2 (2) The conceptual Fig. 1 can be further improved. It is not clear why heterogeneity increases towards the two ends of the three pdfs. Again, this really depends on how heterogeneity is defined. For example, if heterogeneity means the range of a habitat variable, then the three pdfs have

the same heterogeneity (assuming they have the same x range). If heterogeneity means sampling uncertainty, then less skewed pdfs have higher heterogeneity (and uniform distribution has the highest heterogeneity).

Author response We agree and hope that the additional text added due to concern (1) will help here also. We also agree that the range needs to be clarified earlier than the methods, so we have added in the Fig. 1 description that

“Range is calculated as the extent between the 0.025 and 0.975 quartiles”,

which makes the measure of range variable across the three pdfs. This was done to more closely relate to real variables, and the measure used by Allouche *et al.* [27], where the quartile observed in real data would depend on the sample size (as reviewer #2 highlights in point (8)). We have also highlighted in the Fig. 1 description that

“The colours of the heterogeneity arrow heads for panel a correspond to the specific measures of heterogeneity shown in panel b”.

Reviewer #2 Are the lower panels assumed or implied from the pdfs of the upper panels?

Author response We thank the reviewer for highlighting this is not clear in Fig. 2 a-p. We have included in the figure description that the lower panels

“are calculated based on analytically derived Equations for beta and gamma distributed variables, and correspond exactly to the probability density functions (panels a,i)”.

Reviewer #2 I was also very confused with the term “cluster”. I made me several readings to finally realize “cluster” is really “skewness” of the pdfs. Cluster is commonly used to describe spatially explicit heterogeneity, not for study like this one.

Author response We agree that use of the word “clustered” would cause confusion due to its common use in spatial statistics. We have, therefore, changed all instances of the word “clustered” to “concentrated” as an alternative synonym (and alternative antonym to dispersion) that may be less likely to cause confusion (changed on ll. 13, 92, 110 and Fig. 1 description). This alternative is important because, although the distribution does become more skewed, it is specifically the reduced dispersion we refer to, as the distribution could become more skewed while retaining the same dispersion.

Reviewer #2 (3) Line 52: Why “in small isolated environments” ? I don’t see that is necessary for area-heterogeneity tradeoff.

Author response We have removed

(ll. 56) “in small isolated environments”.

Originally we thought the justification of the area-heterogeneity trade-off was only plausible in small environments where niche area is already limited, and in isolated environments where reduced immigration fails to utilise the increased niche dimensionality. There is not adequate data to support this statement, so we removed it.

Reviewer #2 (4) Line 78-80: This statement is misleading. For normal distribution, it is true that change in mean only shifts the distribution along the x axis. But that is not necessarily true for other distributions because mean is dependent on both shape and scale parameters, for example, in the case of Gamma distribution.

Author response We totally agree. We raised that assumption because it is common, not because it is always correct. In the next sentence (ll. 88-90) we state that the assumption is violated with the introduction of a bound (like the gamma distribution). To improve clarity, we restructured the sentence to make it clear sooner that the assumption is often incorrect (ll. 88-90). The two sentences now read:

(ll. 86-90) “However, the use of these measures to assess heterogeneity implies the assumption that changes in the mean simply shift the distribution up or down the number line and have no influence on its scale (statistical dispersion) or shape. This assumption is violated with the introduction of a bounding minimum and/or maximum value for a given variable”.

Reviewer #2 (5) Line 125: The notation of the indicator function in Equation (1) is not clear. There are no values that do not belong to real numbers \mathbb{R} . The correct notation should be $\mathbf{1}_{\mathbb{R}}(x)=1$ if $x < x_{\max}$, otherwise $\mathbf{1}_{\mathbb{R}}(x)=0$.

Author response There are two possible misunderstandings that have occurred, which make us realise the notation needs to be better explained in the manuscript. Firstly, infinity is the upper support of the gamma distribution, but infinity is not in the set of all real numbers (\mathbb{R}). This is precisely what we hoped to capture with the notation, such that if x (the maximum or minimum bound) is infinity or does not exist, then it does not belong to the real numbers and the boundary would have no impact on the dispersion of the distribution. We have included as an example

(ll. 141) “ $\infty \notin \mathbb{R}$ ”

to make this clearer. Secondly, the use of x may be misinterpreted as a variable rather than a placeholder to be used on the minimum or maximum support. To make this clearer, we exchanged x for m in this notation and defined m , modifying the text to be

(ll. 137-142) “ $\mathbf{1}_{\mathbb{R}}(m)$ is an indicator function with \mathbb{R} being the set of all real numbers, m is the lower (*min*) or upper (*max*) bound of the variable, and

$\mathbf{1}_{\mathbb{R}}(m) = 1$ if $m \in \mathbb{R}$, otherwise $\mathbf{1}_{\mathbb{R}}(m) = 0$. Note that any number raised to the zeroth power is equal to one, such that if the lower or upper bound does not exist or is not a real number (*e.g.* $\infty \notin \mathbb{R}$) the boundary has no influence on the dispersion nor δ ".

Reviewer #2 (6) It seems reasonable to use the scale parameter of Gamma distribution to measure heterogeneity but I am concerned about the δ_2 of the Beta distribution (Equation 13). Let me illustrate my concern using the following two numeric cases.

Uniform distribution: When $p=q=1$, Beta distribution is a uniform distribution $U(0,1)$. In my opinion, $U(0,1)$ has the highest heterogeneity than any other Beta distribution (in terms of sampling uncertainty). This leads to $\delta_2=1/3=0.333$.

Let's now set $p=0.5$, $q=1$, Beta distribution becomes skewed to right (*i.e.*, most x values are small, with mode on the smallest x). The heterogeneity of this distribution is lower than the uniform distribution, but $\delta_2=1/(1+0.5+1)=0.4$ is higher than 0.33. This is not consistent to our expectation.

Author response The core of this issue comes from reviewer #2's concern (1), on the definition of heterogeneity, and the fact that we consider both variability (dispersion, inequality, *etc*) and diversity (information entropy) to be different characteristics of heterogeneity that both need to be taken into consideration. Dispersion is the dissimilarity of realisations against the mean, whereas information entropy describes the evenness of the distribution. As a consequence, due to these defined characteristics of dispersion and information entropy, we expect δ_2 to be able to increase beyond its level at $U(0,1)$ where information entropy is maximised (note the same realisation has been shown for Gini = 0.33 already [61]). To illustrate this, we have some numeric cases to show how δ_2 can be interpreted as a mean-independent measure of dispersion for double-bounded variables.

Consider what occurs at the limit when we maximise the variance of a beta distributed variable (X) with a given mean. With a mean of 0.5, maximising the variance for a beta distributed variable would result in an equal probability of observing 0 or 1 and the variance is maximised at 0.25 ($\lim_{q \rightarrow 0^+, p=q} Var(X)$). Meanwhile, with a mean of 0.75, maximising the variance would result in a concentration of probability at the upper bound ($P(1) = 0.75$; $P(0) = 0.25$), and the variance is maximised to 0.1875 ($\lim_{q \rightarrow 0^+, p=3q} Var(X)$). Without a centred mean, it is mathematically impossible for the variance to be maximised to 0.25 for any distribution on $[0,1]$. However, if we calculate δ_2 for each of these distributions, we find they both have a $\delta_2 = 1$: $0.25/(0.5 \cdot 0.5) = 1$; $0.1875/(0.25 \cdot 0.75) = 1$. Both distributions have maximised dispersion for the given mean and thus both have the maximum δ_2 .

If we were to instead set the variance to 1/3 of its maximum for a given mean, we would have $U(0,1)$ when the mean is 0.5 ($p=q=1$), and a variance

of $1/12$. Meanwhile, with a mean of 0.75 ($p = 1.5, q = 0.5$), $1/3$ of the maximum variance is 0.0625 . For both distributions the dispersion is $1/3$ of its maximum (given the constraints of the boundaries) and as a result $\delta_2 = 1/3$ for both distributions.

We hope that the improvements to the manuscript resulting from reviewer #2's concern (1) have helped highlight that there are many distinct characteristics of heterogeneity. Moreover, comparing δ to another characteristic of heterogeneity, such as measurement uncertainty (information entropy), would be hampered by the fact that information entropy is inherently mean biased. This is because $U(0,1)$ must have a mean of 0.5 and any deviation from 0.5 results in greater a skewness and reduced entropy. Developing a mean-independent information entropy will be an important topic of future work (beyond the scope of this article), so we have included a paragraph to convey this:

- (ll. 306-318) “It is also worth noting that there are numerous distinct characteristics of heterogeneity other than dispersion, including information entropy and inequality or skewness, all with important applications. For example, diversity indexes (*e.g.* Shannon diversity index) are often used in ecology to describe the evenness in abundance between each species [44]. For which, even though we have shown that there does exist a positive correlation between dispersion (δ) in foliage heights and the Shannon diversity index for bird species (Extended data Fig. S7b), information entropy is likely a more descriptive characteristic of this specific variable due to it capturing the uniformity or measurement uncertainty of the distribution (Extended data Fig. S7c). Likewise, measures of inequality (or skewness) are often preferred for evaluation of socio-economic outcomes where differences between all individuals are of interest [53]. Disentangling these other characteristics of heterogeneity from the mean (Fig. 2) will be an important topic of future research”.

Reviewer #2 (7) I am not convinced that the numeric examples shown in Fig. 3 and Fig. S2-S5 are credible. Since δ is explicitly included in Equation (50), it of course recovers the linear relationship. A real test should start with Gamma distribution (or Beta distribution) and assumes a function $y=f(x)$ that relates diversity (y) with a habitat variable (x), where x follows a Gamma (Beta) distribution. This approach does not pre-dictate any relationship between diversity and δ but let's the heterogeneity indices speak to see who would do a better job.

Author response We actually used what we think is a similar solution to the one the reviewer suggests, so this comment makes us realise we need to explain it much better. The $f(x)$ that we found in the empirical data determined how Equation (50) was defined. To improve clarity, we have changed

- (ll. 681-683) “set to closely match empirical data” to “set to be equivalent to

model #5 in Table 1 if the independent variables were not scaled to compare effect sizes”

and we included in the Fig. 3 description that $f(x)$ is

“equivalent to model #5 in Table 1”.

The reviewer rightly points out that panels l-n were implicit in the formulation of Equation (50). This was done so that the shape and concentration of the empirical data could be compared. From the reviewer’s comment we see that the additional evidence provided by this was previously overstated in the text. We have, therefore, removed all reference to the theoretical results for δ as an indication of it retrieving true relationships (ll. 223-225, ll. 236-238, and the Fig. 3 description). We have also removed the δ panels for the Extended data Figs. S2-S5. However, we still retain the δ panels in Fig. 3 with a note in the description that

“ δ is implicit in the theoretical species richness model, so panels **m-n** are included only for comparison with the shape and concentration of empirical data”.

This way they should not be interpreted as further evidence of the relationships shown in empirical data, but the shape and concentration of the empirical data can still be compared with the theoretical predictions for all measures. Conversely, the theoretical panels for other heterogeneity measures can be seen as further evidence of the mean-bias shown in empirical data. This is because they are all used as measures of dispersion and are calculated directly from the mean-heterogeneity relationships for gamma distributed variables. The theoretical panels for other heterogeneity measures, therefore, demonstrate that given a system with a linear mean-independent dispersion-diversity relationship, the specific mean bias of current measures results in warped observed relationships.

Reviewer #2 (8) In the section of Line 436 of page 25: In application, the lower bound (min) and upper bound (max) are unknown but subject to sampling intensity. Larger samples will have higher chance to obtain smaller min and larger max. This could be problematic in real application because two samples of different size could lead to very different delta values (I did a quick numeric calculation and saw the delta values could be 2-3 times different for different sample sizes). A solution could be to estimate min and max using order statistics.

Author response We totally agree, and decided to add a warning against deriving bounds from small samples

(ll. 157-160) “where the bounds can be determined conceptually or experimentally. Bounds should not be determined as the observed minimum or maximum from a small sample without conceptual or experimental support”.

Fortunately for most variables, a conceptual determination of bounds is obvious, and no data is required, e.g. trees do not have negative height, and crop cover cannot exist in a proportion greater than 1, *etc.* For some situations, such as phase transitions of materials, the bounds are not obvious. For these situations, experiments can be designed to determine the boundaries much more precisely than sampling, e.g. heating water until the phase change occurs, rather than randomly sampling liquid water and observing the maximum temperature.

Reviewer #2 (9) Line 406: You do not define “general beta and gamma distributions”. Instead, you use the delta formulas derived from the Beta and Gamma distributions.

Author response The reviewer is right that this was not clear originally and that additional information was necessary to retrieve the general pdfs for the beta and gamma distribution. We have, therefore, included an additional paragraph for each distribution:

(ll. 505-509) “The general beta pdf can be expressed using the standard beta pdf ($f(x)$; Equation (6)) as $f_g(v; min, max) = f(\frac{v-min}{max-min})/(max - min)$, where the denominator normalises the pdf to adjust for the modified scale when modelling V , x for indexing the pdf is found for V with Equation (29), and the parameters p and q can be found for V with Equations (30), (33), (10), and (11)”.

(ll. 525-528) “The general gamma pdf can be expressed using the standard gamma pdf ($f(x)$; Equation (20)) as $f_g(v; min) = f(v - min)$, where x to be used for indexing the pdf is found for V with Equation (34), and the parameters k and δ_L can be found for V with Equations (35), (37), and $k = \mu/\delta_L$ ”.

Reviewer #2 (10) I would suggest avoiding using ”axiomatically”. They are not axiomatic. You may not call delta ”true” mean-independent. It is probably not for distributions other than Gamma/Beta.

Author response We have removed all instances of the word axiomatic(ly) (ll. 70, 151, 164, 320, 374, 439, 561, and in the abstract). In its place we have been specific that it is proven for beta and gamma distributed variables. Furthermore, where we say δ is truly mean-independent regarding empirical data we make it specific that it is for

(ll. 194) “these (assessed) variables”.

In the abstract, we state that δ is “essential for understanding true mean-independent heterogeneity relationships in wider research”. This statement is still correct even if some variables are not well modelled by beta or gamma distributions, particularly considering the flexibility of these distributions and the countless variables in numerous disciplines that have been modelled effectively by them.

3 Closing remark

We hope that our responses and changes fully address all concerns and suggestions from the reviewers. If we are mistaken in any instance, we would be grateful for any further feedback to improve the manuscript. Thank you for your time.

Cameron Pellett and Rubén Valbuena

1 General response

We would like to extend our gratitude to both reviewers for their reports thus far. We are glad that we have satisfied all the requests of Reviewer #1. We also greatly appreciate the depth of analysis Reviewer #2 has given in this round of review. Their analysis has highlighted areas where further clarification is needed to prevent erroneous interpretation. The reports of both reviewers have allowed us to make numerous significant improvements to the manuscript and allowed us to demonstrate that the ideas hold up to scrutiny.

We have responded to the individual concerns and suggestions below, where we direct the reviewers to the revised manuscript with the individual line (ll.) changes. Code for replicating simulations and figures is available in the updated zenodo repository for the manuscript (Pellett and Valbuena 2024). References in this report are independent of the manuscript.

2 Response to Reviewer #1

Reviewer #1 The authors did a great revision. All my comments were considered very well. I highly appreciate also the re analyses of the old MacArthur data.

Author response We are very glad to have addressed all of Reviewer #1's comments. Among the many important suggestions Reviewer #1 made, Rapoport's rule was a perfect addition to the manuscript, to both aid readers' understanding and highlight the utility of a measure of heterogeneity that is independent of a variable's boundaries. We thank Reviewer #1 kindly for sharing their ideas and their contribution to this manuscript.

3 Response to Reviewer #2

3.1 δ_2 , mean-independence for beta distributed variables

Reviewer #2 I appreciate very much the effort of the authors in addressing my previous comments. Having read carefully the revision and the methods, I am not yet convinced that the delta is a true heterogeneity measure independent of mean. Take the beta distribution as an example, neither equation (9), (13) nor (33) proves that δ_2 is independent of mean. Opposite to that claim, it is rather straightforward to show that they are not. Equation (9) or (13) is dependent on both p and q . Although μ does not appear in this equation, μ is a function of p and q ($\mu=p/(p+q)$). As such, equation (13) can be equally written as: $\delta_2=\mu/(p+\mu)$. Keep in mind, p and q independently vary. This relationship means given a fixed p , δ_2 asymptotically increases. This is easy to show numerically. Given a p , we can change q , e.g., randomly sample from a uniform distribution. I wrote a simple R program to do that.

```

function (pfixed,nsample) {
# Test the relationship between delta2 and mean of beta distribution
# For a given p value, randomly sample q from a uniform distribution
# plot delta2 versus mean of the beta distribution
# Here I also plot beta variance versus mean
# If the author's claim were true, we would expect a flat line between
delta2 and beta mean.

beta.mean=numeric()
beta.var=numeric()
beta.delta2=numeric()

p=pfixed
q=runif(nsample,0,10)
beta.mean=p/(p+q)
beta.delta2=1/(1+p+q)
beta.var=p*q/((p+q)^2*(p+q+1))

par(mfrow=c(1,2))
plot(beta.mean,beta.var)
curve(x^2*(1-x)/(p+x),add=T,lwd=2,col='red')
plot(beta.mean,beta.delta2)
curve(x/(p+x),add=T,lwd=2,col='red')
}

```

The following graphs are produced by running the above program by assigning `pfixed=2` and `nsample=1000`. The left panel shows the relationship between mean and variance of beta distribution. Yes, they are not independent as the authors recognize. The red curve is the analytical variance-mean relationship: $\text{var}=\mu^2(1-\mu)/(p+\mu)$ which perfectly matches the black circles calculated from the 1000 samples of q from uniform distribution `runif(0, 10)`.

The right panel shows the relationship between `delta2` and the mean of beta distribution. Clearly, they are not independent at all. `Delta2` is perfectly described by the analytical formula (the red curve): $\text{delta2}=1/(p+q+1)=\mu/(p+\mu)$.

No matter what p values are used, the relationship between delta2 and mean are not independent.

Author response This criticism has highlighted clear improvements that are required for understanding the manuscript: namely, an improved definition and demonstration of what a scale parameter is and how it can be interpreted. The bottom line being that fixing one of the beta distribution’s shape parameters (p or q) and changing the other will change both the mean and scale of the distribution. We will describe the changes incorporated into the manuscript to prevent this misunderstanding momentarily, but first it is important to highlight the issue with Reviewer #2’s argument by exploring the simulation they ran.

Reviewer #2 decided to fix p and select q between 0 and 10. The argument was that a measure of mean-independent heterogeneity should remain fixed for all of the resulting distributions. However, this is certainly not the case: if there was a mean-independent measure of heterogeneity, it should change for all of the distributions in the simulation. This can be made clear with just two examples of the distributions Reviewer #2 used. The first beta distribution (B_1) with $q = 4$ and the second beta distribution (B_2) with $q = 1$. Both with $p = 2$ as fixed by Reviewer #2. The mean of B_1 is $\mu = 1/3$ and the mean of B_2 is $\mu = 2/3$. Reviewer #2’s argument is that both of these distributions differ only by mean and that the heterogeneity is identical, otherwise a measure of mean-independent heterogeneity would be expected to change. Now consider the density functions of B_1 and B_2 in Fig. R1. For both distributions, the means’ symmetric distances to the boundaries are the same and thus should not influence the symmetric shape or scale of the distribution. However, it is clear that the two distributions have vastly different levels of heterogeneity and statistical dispersion. B_1 is significantly more concentrated at the mean than B_2 . Therefore, we would expect a mean-independent measure of heterogeneity to increase from B_1 to B_2 , where δ_2 is performing as expected in Reviewer #2’s simulation. The fundamental problem with

Reviewer #2's argument being that fixing p and changing q modifies both the mean and the scale of the distribution, not just the mean.

Figure R1: Fixing p and changing q modifies both the mean and scale of the beta distribution. Probability density functions of two beta distributions (B_1 and B_2) with one fixed parameter $p = 2$ and q selected to give $\mu = 1/3$ (B_1 : $q = 4$) and $\mu = 2/3$ (B_2 : $q = 1$). The dashed blue line is the mean (μ).

To clarify our point further we will rerun the simulation, except this time we fix the precision parameter (ϕ) of the beta distribution as introduced in 2004 by Ferrari and Cribari-Neto. ϕ is a good parameter to fix because it was derived independently of the work we present in the manuscript and it is well established when modelling beta distributed variables (see the betareg r package; and paper by Cribari-Neto and Zeileis 2010). Ferrari and Cribari-Neto (2004) define $\phi = p + q$. The mean (μ) can, therefore, be changed by diverging p and q while keeping their sum fixed. This gives $p = \mu\phi$ and $q = (1 - \mu)\phi$. B_1 and B_2 are now redefined with the same precision parameter $\phi = 6$, and $\mu = 1/3$ (B_1) and $\mu = 2/3$ (B_2), respectively. The result in Fig. R2 shows the two density functions are equally concentrated around the mean, and we would expect a mean-independent measure of heterogeneity to give the same value for both distributions. It should come as no surprise that $\delta = 1/7$ for both

distributions.

Figure R2: Fixing the precision parameter (θ) maintains the scale of the distribution with changing mean (μ). The mean-independent measure of heterogeneity (δ) is, therefore, also fixed with a changing mean.

The fundamental misunderstanding of the effect of the shape (p, q) or scale (δ_2) parameters on the beta distribution seems to have arose due to the previous version of our manuscript lacking a clear definition of the scale parameter. We corrected this mistake by adding the brief definition, equivalent to the definition given by Ferrari and Cribari-Neto (2004):

- (lls. 330-332) “A parameter is interpreted as a scale/dispersion parameter when, for a constant mean, increases in the parameter always correspond to increases in the variance...”.

Following this it should be clear that the shape parameters (p and q) of the beta distribution will not meet the requirements of a scale parameter and neither can be fixed in an attempt to fix the scale of the distribution.

Reviewer #2 What happens if both p and q are randomly sampled from uniform distribution (or any other distribution)? Again, it is easy to show that δ_2 and mean of beta distribution are not independent. The following program shows that.

```

function (nsample) {
# Test the relationship between delta2 and mean of beta distributions
# (1) Sample p and q each from uniform distribution unif(0,10)
# (2) Calculate mean (=p/(p+q)) and delta2 (=1/(1+p+q)) for each
sample
# (3) Plot the relationship between delta and mean.
# If delta2 is really independent of mean, the relationship should be a flat
line.

beta.mean=numeric()
beta.var=numeric()
beta.delta2=numeric()

p=runif(nsample,0,10)
q=runif(nsample,0,10)
beta.mean=p/(p+q)
beta.delta2=1/(1+p+q)
beta.var=p*q/((p+q)^2*(p+q+1))

par(mfrow=c(1,2))
plot(beta.mean,beta.var)
lines(lowess(beta.mean,beta.var,f=1/3),lwd=2,col='red')
plot(beta.mean,beta.delta2)
lines(lowess(beta.mean,beta.delta2,f=1/3),lwd=2,col='red')
}

```

I ran the program by setting nsample=1000 and obtained the following results which show delta2 and mean are not independent (the right panel). The dependence is no less than the original variance of beta distribution on mean (the left panel). Note the red curves are the local weighted regression lines.

Author response This assessment has again revealed important improvements essential for the manuscript: namely a clear indication that the domain of both the shape parameters, p and q , are on $(0, \infty)$. We will describe the changes made, but first we highlight that correctly sampling p and q on their domain completely corrects the artefacts of mean-bias in δ_2 .

In Reviewer #2's second simulation they randomly sampled p and q from uniform distributions on the domain $(0, 10)$. As a result, p or q were only selected on an small subset of their true domain and neither could be selected greater than 10. The small domain used in sampling is the reason for the observed white space in Reviewer #2's figure for the variance (σ^2) and δ_2 from their simulation. Consider just one example where $\delta_2 = 0.05$ and $\mu = 0.1$. For this to be realised, the shape parameters must be $p = 1.9$ and $q = 17.1$. This is entirely possible for the beta distribution, but it is not possible in the simulation defined by Reviewer #2. Meanwhile, the same $\delta_2 = 0.05$ is achievable with $\mu = 0.5$, with $p = 9.5$ and $q = 9.5$. It is this censoring of the vast majority of possible beta distributions that results in the dependence observed between δ_2 and the mean.

To further clarify this point we can rerun the simulation, except this time we randomly sample p and q from a distribution that is on their true domain, $(0, \infty)$. Here we sample the exponential distribution ($\text{Exp}(\lambda)$) with rate parameter $\lambda = 10$. An exponentially distributed random variable ($X \sim \text{Exp}(\lambda)$) is on the domain $(0, \infty)$. We reran the simulation taking $n = 10^6$ independent random samples for p and q from $\text{Exp}(\lambda)$. Now consider Fig. R3. Without the censoring implicit in sampling p and q from a small subset of their true domain, δ_2 is observed to be independent of the mean, matching empirical data. Additionally, the variance is also observed to exhibit the same mean-dependence described theoretically and empirically in the manuscript.

Figure R3: Randomly sampling shape parameters p and q on the domain $(0, \infty)$ from an exponential distribution on the same domain with rate parameter $\lambda = 10$. $n = 10^6$.

We are grateful to Reviewer #2 for indirectly highlighting that we did not specify the domain of the distribution’s parameters in the text. This is important to ensure readers do not misunderstand the distributions used. To correct this we have now specified the domains of the parameters p and q for the beta distribution.

(ll. 342) “... p and q both on the open interval $(0, \infty)$...”.

Additionally, we have been more precise about whether parameters are on the closed or open interval (lls. 343, 352, 355).

It is still important to note that sampling p and q is not a good method of evaluating the possible beta distributions that could arise and the relationship between the mean and scale. A better solution would be to sample means from a uniform distribution on $(0,1)$, then for each sampled mean, sample the variance from a uniform distribution on $(0, \max_{\mu}\{\sigma^2\})$. Heterogeneity measures, including δ_2 , can then be calculated. Carrying out this simulation yields the results in Fig. R4, which make the limits of possible values and levels of concentration much clearer. Note that the maximum variance ($\max_{\mu}\{\sigma^2\}$) is dependent on μ as described in the previous round of review (response to Reviewer #2’s point 6). We now include a proof (using L’Hôpital’s rule) of this fact in the new Supplementary Information (lls. SI 68-84), which is referenced in the methods for the beta distribution with

(lls. 366-368) “This dependence between the variance and the mean is most apparent at the variance’s maximum limit when $p, q \rightarrow 0$, and the variance is equal to $\mu(1 - \mu)$ (see Supplementary Information SI2)”.

Figure R4: Simulation demonstrating the extent of possible beta distributions. The simulation was carried out by selecting the mean (μ) as random samples of a uniform distribution on the domain $(0, 1)$. Then subsequently sampling the variance (σ^2) for each μ from additional uniform distributions on the domain $(0, \max_{\mu}\{\sigma^2\})$. Where $\max_{\mu}\{\sigma^2\} = \mu(1 - \mu)$ is the maximum variance possible for a given mean, see proof in Supplementary Information SI2. $n = 10^6$

We also realise that Reviewer #2 may have seen their formulation of

$$\delta_2 = \frac{\mu}{p + \mu}, \text{ and}$$

$$\sigma^2 = \frac{\mu^2(1 - \mu)}{p + \mu},$$

as alone proof that δ_2 is mean biased. However, it can be made clear that this is not the case by demonstrating the need for simplification when expanded. Reviewer #2's formulation relies on the equality, $(p + q + 1)\mu = p + \mu$ (see Appendix at the end of this report if not immediately obvious). By this equality a redundant μ/μ can be embedded in the expression for δ and σ^2 :

$$\delta = \frac{1}{p + q + 1} = \frac{1}{p + q + 1} \cdot \frac{\mu}{\mu} = \frac{\mu}{p + \mu}, \text{ and}$$

$$\sigma^2 = \frac{pq}{(p + q)^2(p + q + 1)} = \frac{\mu(1 - \mu)}{p + q + 1} = \frac{\mu(1 - \mu)}{p + q + 1} \cdot \frac{\mu}{\mu} = \frac{\mu^2(1 - \mu)}{p + \mu}$$

Note that even without the redundant μ/μ in the expression for σ^2 , a component of σ^2 is still expressed with μ , no matter our attempts to further simplify. This mean dependence ($\mu(1 - \mu)$) is the same dependence found at the maximum limit of σ^2 as shown in the additional proof of Supplementary Information SI2 (equation (56)). Conversely, when we simplify the expression for δ_2 and the redundant μ/μ is removed, we find precisely the component of σ^2 that cannot be expressed with μ . This is what δ is: it is the variance with the components that can be expressed with the mean removed. This is what is proved algebraically in the methods of the main text for both the beta and gamma distributions.

We are extremely grateful to Reviewer #2 for making clear the need for further clarification and demonstration of the parameters of the beta distribution. We think that after this explanation and the additional clarifications included in the revised manuscript, it is now undoubtable that δ_2 is a mean-independent measure of heterogeneity for beta distributed variables. That said, and following both rounds of review, we have decided to make it clear earlier in the manuscript that the theoretical demonstration of δ is for the beta and gamma distributions. We add this clarification to the abstract:

“for beta and gamma distributed variables”.

We also add further clarifications of the distributions used in the main text on lls. 70, 129, 138, 157 and 637. Finally, we also adjust the text to be more accurate, namely with empirical data providing support not proof:

- (ll. 69) “While we can support its mean-independence with any approximately beta or gamma distributed variable, we”.

3.2 δ_L , mean-independence for gamma distributed variables

Reviewer #2 The above are the examinations for the relationships between variance and mean of beta distribution and also the relationship between δ_L and mean. It can also be shown that for other cases, including the low bounded Gamma distribution, equation (3) in the main text does not remove the effect of mean either. I see it can reduce the effect of mean but not remove it. It is not hard to understand why this is the case. Let's put the problem in a more general situation by assuming that the variance-mean follows Taylor's power-law: $\text{var} = a \cdot \mu^b$. By dividing var by μ , it deduces to $\text{var} = a \cdot \mu^{(b-1)}$. In the case of Poisson distribution, this perfectly removes the effect of μ but for any other cases, this does not remove the effect of μ . Most empirical data show b ranges between 1 and 2. Thus dividing variance by μ will not remove the effect of mean, regardless whether we consider the bounds or not. This may also explain why there is still a pronounced trend between δ_L and mean in Fig. 2t as observed also by the other reviewer. This trend is not a "negligible trend" as the authors argue but is inherent.

Author response We greatly appreciate Reviewer #2's suggestion of Taylor's power-law. We suspect the law will be key to extending the ideas of this manuscript to discrete, count variables. However, we find it to be incorrect in Reviewer #2's specific use, and we disagree that Reviewer #2's analysis of the beta distribution could be mirrored for the gamma distribution. We will detail the latter issues first, then discuss possible extensions.

Reviewer #2 states that, as in their analysis of the beta distribution, it can also be shown that equation (3) does not remove the effect of mean for gamma distributed variables. The gamma distribution is simpler than the beta distribution because it already has a scale parameter without needing to be reparametrised. Therefore, there are only two parameters that could be fixed when evaluating the distribution. As we have already described for the beta distribution, the obvious parameter to fix when assessing mean-bias of heterogeneity measures is the scale (or precision) parameter. Fixing the scale parameter is exactly what was done in the manuscript. However, if we ignore the analysis of the beta distribution done in this review and attempt to fix the shape parameter (k), we see similar problems. Consider Fig. S1 in the manuscript, where the coefficient of variation (CV) is fixed and the mean is, as a result, changed entirely by how dispersed the distribution is (relying on the lower boundary). This figure is actually equivalent to fixing k and modifying δ_L to change the mean (note equation (24), $\text{CV} = 1/\sqrt{k}$). However, like the beta distribution, it is clear from the density functions that the heterogeneity of the distributions is not constant and would be expected to change with the mean. To make this conclusion clear to readers we have added some further clarification to Fig. S1's description.

Regarding the use of Taylor’s power-law, it is first important to note that the law is not a general situation for bounded continuous variables. All of the evidence we have seen for Taylor’s power-law is for discrete, count variables, which for applications in ecology are primarily species abundances. The natural null model for these variables is the Poisson distribution where, in an idealised empirical example, positions of individuals of a single species are independent and random and the mean and variance are identical ($\sigma^2 = \mu$). Of course, for most species, individuals are not independent and randomly positioned due to birth, cooperation, and competition. Taylor’s power-law defined as $\sigma^2 = a\mu^b$ is, therefore, useful for modelling the mean-variance relationship of these species, with a and b being positive constants for each species. Taylor indicated that b is interpreted as an index of spatial (or temporal) aggregation of individuals, with $b < 1$ indicating greater regularity in the distribution, $b = 1$ indicating the position of individuals is independent and random (Poisson point process), and $b > 1$ indicating greater clustering or aggregation of individuals. The purpose of a and b is, thus, akin to the purpose of δ but for vastly different variables, where the mean-variance relationships are also expected to be vastly different. This is quite obvious when comparing the mean-variance relationships observed for the bounded continuous variables in this manuscript (Fig. 2) and the mean-variance relationships observed for the discrete count variables in Taylor’s seminal paper and subsequent work (Taylor 1961; Taylor, Woiwod, and Perry 1978). For Taylor’s species abundance data the observed mean-variance relationship resembles a single curve with constant parameters for each species, where variability around the curve is generated through sampling uncertainty alone. Meanwhile, for the bounded continuous variables in this manuscript, the mean-variance relationship resembles a continuum, not a single curve, with variability around the general trend being a characteristic of the latent variability of each distribution, not measurement uncertainty. It is for this reason that the mean-variance relationships of bounded continuous variables is described by a nonconstant scale parameter. For gamma distributed variables this is $\sigma^2 = \delta_L\mu$, where δ_L can vary for each study site and resulting distribution, and is not a constant for the variable like a or b in Taylor’s power-law. Note that $\sigma^2 = \delta_L(\mu - min)$ allows for the description of the entire continuum with varying δ_L (see Fig. 2e). The same conclusions can be made for beta distributed variables with $\sigma^2 = \delta_2([\mu - min][max - \mu])$ (see Fig. 2m).

If we ignore our disagreement with the justification of Taylor’s power-law as a ‘general’ situation, we can still evaluate its ability to model mean-variance relationships for single gamma or beta distributions. Reviewer #2 noted that dividing σ^2 by μ would perfectly remove the effect of μ for the Poisson distribution. This is true assuming Taylor’s power-law with just one set of constants: $a = b = 1$. Additionally, though not mentioned by Reviewer #2, dividing σ^2 by μ would also perfectly remove the effect

of μ for a gamma distribution. This is true assuming Taylor’s power-law with a different set of constants: $a = \delta_L$, and $b = 1$. To demonstrate this, consider the variance of a gamma distributed variable, given in equation (22) in the manuscript and copied here:

$$\sigma^2 = \delta_L \mu = \delta_L \mu^1,$$

and note the exact equivalence to Taylor’s power-law when $b = 1$ and $a = \delta_L$:

$$\sigma^2 = a\mu^b, \quad \text{if } a = \delta_L, b = 1.$$

For this relationship, dividing σ^2 by μ gives

$$\frac{\sigma^2}{\mu} = a\mu^{b-1} = a\mu^0 = a, \quad \text{if } b = 1,$$

and for gamma distributed variables $b = 1$ and $a = \delta_L$. This is also given in the manuscript with equation (23). Meanwhile, Taylor’s power-law is completely unable to model the mean-variance relationship for beta distributed variables. This is because, as defined, $a\mu^b$ is a continually increasing expression, which is unable to model a hump-shaped mean-variance relationship. Note that this is not evidence against the beta distribution’s mean-variance relationship, it is simply a limitation of the expression, $a\mu^b$. In the same way that the fact that $\sin(x)$ is an oscillating function is not disproven due to x^2 being a continually increasing function. Likewise, the fact that $a\mu^b$ cannot be used to model double-bounded variables is not evidence against Taylor’s power-law because the law was defined for discrete count variables, not bounded continuous variables.

For these reasons, we do not agree that Taylor’s power-law gives any evidence of an inherent trend between σ^2/μ and μ for gamma distributed variables or the lower-bounded land elevation above sea level. It is also important to note that even if there was a mean- δ relationship remaining, it is not significant in the sense that it does not change the final conclusions of the manuscript: δ does reveal a monotonic linear heterogeneity-diversity relationship in data previously used to support theories of a hump-shaped relationship. Furthermore, δ only corrects for mean-dependence arising from the boundaries of the variable, which allows for the evaluation of hypotheses like Rapoport’s rule (as suggested by Reviewer #1) where mean-dependence may arise from ecological effects outside of the mathematical constraints of the variable.

With the above disagreements stated, we still greatly appreciate Reviewer #2 bringing Taylor’s power-law to our attention in this context. We appreciate it because, as Reviewer #1 also suggested, extending the ideas presented in this manuscript to discrete count variables would be

an important contribution. Taylor’s power-law may indeed be the key to this. However, we think this extension or unification is deserving of its own manuscript, and we think its inclusion in this manuscript would require a great deal of background that would distract from the main focus of this work.

3.3 Practical guidelines for determining boundaries

Reviewer #2 My another criticism is that the study is entirely taciturn about how one may determine whether a set of empirical data are low bounded, upper bounded, double bounded or unbounded. For example, I collect a set of data on soil nutrients or climatic temperatures. Should I treat them as low bounded, upper bounded, or unbounded at all? It is intuitive to consider elevation as bounded data because elevation has physical boundaries but deciding bounds for other types of data is much more elusive. Should we just use equation (29) to transform any data into double bounded data? Practical guidelines are needed for empirical data analysis.

Author Response We understand now that the determination of boundaries requires further attention, especially considering its correct interpretation is fundamental to the manuscripts ideas. Confusion between an empirically observed minimum or maximum value and a variable’s lower and upper boundary seem to be the primary issue. To prevent this confusion we have made numerous changes to the manuscript. Firstly, we think this misunderstanding has partially arose due to the notation used for the lower and upper boundary in the manuscript: *min* and *max* are common as function names in programming languages like R for taking the empirical minimum or maximum from a given dataset. We have, therefore, replaced all notation of *min* and *max* with L and U , respectively, in hopes of avoiding this incorrect association. Additionally, we have replaced the use of minimum and maximum when describing theoretical boundaries where we think it could be misunderstood. These changes were made in the text on lls. 11, 88, 107, 111, 113, 130, 132, 140, 146, 149, 151, 485, 486, 489, 498, 506, 511, 526, 532, 534, 537, 542, and 554; in equations (1), (2), (3), (4), (29), (30), (31), (32), (33), (34), (35), (37), (38), (39), and (40); and in Fig. 2. Secondly, as suggested by Reviewer #2, we have included additional guidance about identifying boundaries, for which, we think examples are the best way to develop intuition and correct understanding. To this end, we have briefly stated four additional examples in the main text with

lls. 161-164 “Examples of potential boundaries include the proportion of land covered by forest ($L = 0$ and $U = 1$), animal mass ($L = 0$ g), beetle size limited by oxygen ($L = 0$ cm and $U = 16$ cm), and biochemical fish depth limits ($L = 0$ m and $U = 8, 200$ m)”.

Additionally, we have added a section to the Supplementary Information (referred to on lls. 164 and 639), where we give additional details, pro-

vide examples from numerous fields, and impart general guidelines about variables that could be considered more elusive (see Supplementary Information SI1; lls. SI 1-50).

We hope these examples and additional information will answer Reviewer #2's questions. However, should they not, we also reply directly to the questions here. The mass of soil nutrients can be considered a lower bounded variable, where the mass of nutrients will not be less than zero. Climate temperature is discussed in the supplementary information as an example with insignificant boundaries when measured on Earth, with the lower bound at -273.15°C being so distant from the expected distribution it can be disregarded. The final point is that equation (29) certainly cannot be used to 'transform' a variable into a double-bounded variable. We think this arose due to the requirement of more examples and guidance about the distinction between the boundaries and the empirical minimum and maximum value, which we have now added. We think these changes have improved clarity and we appreciate Reviewer #2 highlighting this need.

3.4 Truncated variables

Reviewer #2 A minor point regarding equation (38): Couldn't it make more sense to use truncated distribution for upper bounded data? This is because "max" is not observable in empirical data as I indicated in the 1st round of review. I believe truncation offers a solution.

Author Response We are grateful to Reviewer #2 for bringing up truncated distributions as they may be a very important consideration for other applications. We now give further details of truncated variables in Supplementary Information SI1:

lls. SI 51-67 "Truncated variables are a distinct category from those described above and in the main text. Truncated variables primarily arise due to measurement limitations, where individuals or units below or above a threshold do exist but are not observed. For example, foresters and forest ecologists often measure tree diameters at a fixed height of 1.3 meters above the ground. As a result, the diameter distribution has a distinct end point greater than the true boundary at zero. This arises because trees with smaller diameters than the observed end points are not tall enough to be measured. As a result, a distribution's mean can be moved towards the measurement threshold and a greater number of units will simply no longer be observed. In this case there may be little to no concentration at the threshold, assuming the true boundary is far from the measurement threshold. We consider this to be a measurement problem specific to individual fields of study, so it is not possible to give universal solutions. It may be necessary and possible in some instances to model the un-

measured observations, such as assuming an underlying parametric distribution and sampling the remaining observations. However, this must be carried out with caution and often solutions to the underlying measurement problem may be necessary”.

We hope that the changes and additions regarding the identification of boundaries clarifies that the “*max*” (now denoted U) can be observed in empirical data or can at least be determined conceptually or experimentally. Examples of this include the proportion of crop cover and replication rate of mammals to name a few (see the newly added Supplementary Information SII1 for details). Additionally, we think that one example in the previous version of the manuscript given for materials in specific states could have created additional confusion. The temperature thresholds between liquids, solids, and gases are not good examples of boundaries and are likely better described with truncated distributions. We are grateful that Reviewer #2 highlighted this issue, and the current version of the manuscript has it removed (lls. 644-646).

3.5 Closing remarks

Reviewer #2 I consider this is an interesting attempt in addressing a very important problem in ecology. I am just not sure how we may proceed with this study. I don't think I can support the publication of the current form given my concern that the delta does not really remove the effect of mean from heterogeneity (delta). I would like to see further response of the authors. Hope my comments are of use for the authors to further think of the problem.

Author Response We have found Reviewer #2's comments to be exceptionally useful for further considering the problem. We think that the comments and elicited changes will greatly improve readers' understanding and subsequent value of the manuscript. We also greatly appreciate the attempts to demonstrate mean-bias in δ as we think, in addition to helping us improve the manuscripts clarity, it has given us the opportunity to demonstrate the ideas hold up to scrutiny. We thank Reviewer #2 kindly for sharing their ideas and their contribution to this manuscript thus far.

Cameron Pellett and Rubén Valbuena

References

- Cribari-Neto, Francisco and Achim Zeileis (Apr. 2010). “Beta Regression in R”. en. In: *Journal of Statistical Software* 34, pp. 1–24. ISSN: 1548-7660. DOI: 10.18637/jss.v034.i02. URL: <https://doi.org/10.18637/jss.v034.i02> (visited on 05/06/2025).
- Ferrari, Silvia and Francisco Cribari-Neto (Aug. 2004). “Beta Regression for Modelling Rates and Proportions”. In: *Journal of Applied Statistics* 31.7. <https://doi.org/10.1080/0266476042000214501>, pp. 799–815. ISSN: 0266-4763. DOI: 10.1080/0266476042000214501. URL: <https://doi.org/10.1080/0266476042000214501> (visited on 12/22/2022).
- Pellett, Cameron and Rubén Valbuena (2024). *Data and code - Disentangling dispersion from mean reveals true heterogeneity-diversity relationships*. en. <https://doi.org/10.5281/zenodo.11561448>. DOI: 10.5281/zenodo.11561448. URL: <https://zenodo.org/uploads/11561448> (visited on 06/11/2024).
- Taylor, L. R. (Mar. 1961). “Aggregation, Variance and the Mean”. en. In: *Nature* 189.4766. Publisher: Nature Publishing Group, pp. 732–735. ISSN: 1476-4687. DOI: 10.1038/189732a0. URL: <https://www.nature.com/articles/189732a0> (visited on 05/05/2025).
- Taylor, L. R., I. P. Woiwod, and J. N. Perry (1978). “The Density-Dependence of Spatial Behaviour and the Rarity of Randomness”. In: *Journal of Animal Ecology* 47.2. Publisher: [Wiley, British Ecological Society], pp. 383–406. ISSN: 0021-8790. DOI: 10.2307/3790. URL: <https://www.jstor.org/stable/3790> (visited on 05/05/2025).

4 Appendix

Here we show workings for the equality, $(p + q + 1)\mu = p + \mu$.

$$\begin{aligned}(p + q + 1)\mu &= (p + q + 1)\frac{p}{p + q} \\ &= \frac{p^2}{p + q} + \frac{pq}{p + q} + \frac{p}{p + q} \\ &= \frac{p^2 + pq}{p + q} + \mu \\ &= \frac{p(p + q)}{p + q} + \mu \\ &= p + \mu\end{aligned}$$

1 Response to Reviewer #2

Reviewer #2 I thank the authors for the further clarifications. They help. The issue here is not the definition of the shape parameters of beta distribution which themselves are clear to any readers but that the ranges of the shape parameters have to be large enough. In reality, there is no theoretical basis that p and q ranges have to be large or small, nor p and q should be constrained by $p+q=\phi$. I would like the authors further to clarify that in the ms. Otherwise, I congratulate the authors for the neat contribution that is long needed.

Author response We are very grateful for the further feedback. Reviewer #2 is correct that there is no theoretical basis to define arbitrary (small or large) ranges of shape parameters when assessing possible beta distributions. Instead the approach must be to allow for all possible beta distributions by allowing for all possible shape parameters defined for the beta distribution, *i.e.* $p, q \in (0, \infty)$. We also agree that the selection of possible ϕ or δ_2 should never be allowed to constrain the possible shape parameters (p, q) . To ensure $p, q \in (0, \infty)$, the interval of dispersion/precision parameters can be defined based on the interval of the possible shape parameters. To make this clear in the manuscript we clarify why $\delta_2 \in (0, 1)$ with

lls. 352-354 “The interval constraining δ_2 is found with limits of the interval of p and q as $\lim_{p \rightarrow \infty, q=(1/\mu-1)p} \delta_2 = 0$ and $\lim_{p \rightarrow 0, q=(1/\mu-1)p} \delta_2 = 1$.”

The equivalent for ϕ , discussed in this peer review, can be given as

$$\begin{aligned} \lim_{p \rightarrow \infty, q=(1/\mu-1)p} p + q &= \phi = \infty \\ \lim_{p \rightarrow 0, q=(1/\mu-1)p} p + q &= \phi = 0 \end{aligned} .$$

Additionally, we clarify why $\mu \in (0, 1)$ with

lls. 347-349 “The interval constraining μ is found with limits of the interval of p and q as $\lim_{p \rightarrow 0, q \in (0, \infty)} \mu = 0$ and $\lim_{q \rightarrow 0, p \in (0, \infty)} \mu = 1$.”

Therefore, using μ and δ_2 (or ϕ) to define beta distributions does not constrain p and q nor does it constrain the possible beta distributions. We also state this in the manuscript with

lls. 357-358 “It can, thus, be shown that the combination of μ and δ_2 are capable of describing all possible values of p and q , and by extension all possible beta distributions...”

We are grateful to Reviewer #2 for highlighting the need to clarify this in the manuscript.